# Abstract representations emerge naturally in neural networks trained to perform multiple tasks

W. Jeffrey Johnston [1,2] ✉ & Stefano Fusi [1,2] ✉

Humans and other animals demonstrate a remarkable ability to generalize knowledge across distinct contexts and objects during natural behavior. We posit that this ability to generalize arises from a specific representational geometry, that we call abstract and that is referred to as disentangled in machine learning. These abstract representations have been observed in recent neurophysiological studies. However, it is unknown how they emerge. Here, using feedforward neural networks, we demonstrate that the learning of multiple tasks causes abstract representations to emerge, using both supervised and reinforcement learning. We show that these abstract representations enable few-sample learning and reliable generalization on novel tasks. We conclude that abstract representations of sensory and cognitive variables may emerge from the multiple behaviors that animals exhibit in the natural world, and, as a consequence, could be pervasive in high-level brain regions. We also make several specific predictions about which variables will be represented abstractly.

The ability to generalize existing knowledge to novel stimuli or situations is essential to complex, rapid, and accurate behavior. As an example, when shopping for produce, humans make many different decisions about whether or not different pieces of produce are ripe—and, consequently, whether to purchase them. The knowledge we use in the store is often learned from experience with that fruit at home—thus, generalizing across distinct contexts. Further, the knowledge that we apply to a fruit that we buy for the first time might be derived from similar fruits—generalizing, for instance, from an apple to a pear. The determinations themselves are often multi-dimensional and multi-sensory: both firmness and appearance are important for deciding whether an avocado is the right level of ripeness. Yet, at the end of this complex process, we make a binary decision about each piece of fruit: we add it to our cart, or do not—and get feedback later about whether that was the right decision. This produce shopping example is not unique. Humans and other animals exhibit an impressive ability to generalize across contexts and between different objects in many situations.

The representational geometry of sensory and cognitive variables in a population of neurons provides insight into the computations that the representation may and may not facilitate[1–3]. We hypothesize that the ability to generalize described above is tied to this representational geometry. For instance, neural representations of sensory and cognitive variables are often nonlinearly mixed together. As a result, these representations have high-embedding dimension[4–6]. While this kind of nonlinear dimensionality expansion allows flexible learning of new behaviors[5] and provides metabolically efficient and reliable representations[7], the resulting representation often does not permit generalization across contexts or stimuli[5,8]. Alternatively, factorized, or even linear, representations of the relevant sensory or cognitive variables (i.e., representations that have no nonlinear mixing) often permit this generalization. Recent experimental work has shown that this kind of factorized—and approximately linear—representation exists at the apex of the primate ventral visual stream, for faces in inferotemporal cortex[9–11]. Further, experimental work in the hippocampus and prefrontal cortex has shown that representations of the sensory and

[1]Center for Theoretical Neuroscience, Columbia University, New York, NY, USA. [2]Mortimer B. Zuckerman Mind, Brain and Behavior Institute, Columbia University, New York, NY, USA. ✉e-mail: wjeffreyjohnston@gmail.com; sf2237@columbia.edu

cognitive features related to a complex cognitive task, also support generalization[8]. We refer to representations of task-relevant sensory and cognitive variables that support generalization−like in these examples and others[12–16]−as abstract representations.

In the machine learning literature, abstract representations are often referred to as factorized[17] or disentangled[10,17–20] representations of interpretable stimulus features. Deep learning has been used to produce abstract representations primarily in the form of unsupervised generative models[18,21,22] (but see ref. [23]). In this context, abstract representations are desirable because they allow potentially novel examples of existing stimulus classes to be produced by linear interpolation in the abstract representation space (for example, starting at a known exemplar and changing its orientation by moving linearly along a dimension in the abstract representation space that is known to correspond to orientation)[18].

Here, we ask how abstract representations−like those observed in higher brain regions[8,9]−can be constructed from the nonlinear and high-dimensional representations observed in early sensory areas[6,24–28]. To study this, we begin by mirroring these high-dimensional and nonlinear representations in a learned model of continuous latent variables; then, we show that training feedforward neural network models to perform multiple distinct classification tasks on these latent variables induces abstract representations in a wide variety of conditions.

Experimental work on animals performing more than a couple of distinct behavioral tasks remains nearly nonexistent[29]. However, modeling work using recurrent neural networks has shown that the networks often develop representations that can be reused across distinct, but related tasks[30–32]−though the abstractness of these reusable representations was not measured. Thus, the behavioral constraint of multi-tasking may encourage the learning of abstract representations of stimulus features that are relevant to multiple tasks. To investigate this hypothesis, we train feedforward neural network models to perform multiple distinct tasks on a common stimulus space. Previous work in machine learning has shown that similar multi-tasking networks can achieve lower loss from the same number of samples than networks trained independently on each task[33] (and see ref. [34]), and that they can quickly learn novel, but related, tasks that are introduced after training[35]. Both of these properties are hallmarks of abstract representations−however, to our knowledge, the representational geometry developed by these multi-tasking networks has not been characterized.

We begin by introducing the multi-tasking model and show that it produces fully abstract representations that are surprisingly robust to heterogeneity and context dependence in the learned tasks. These representations also emerge in the more realistic case in which only a fraction of tasks are closely related to the latent variables, and the remaining larger fraction is not. Next, we characterize how the level of abstraction depends on nonlinear curvature in the classification task boundaries and on different types of inputs, including images. We also show that the multi-tasking model learns similarly abstract representations when trained using reinforcement learning. Finally, we use this framework to make several predictions for how neural representations in the brain will be shaped by behavioral demands. Overall, our work shows that abstract representations−similar to those observed in the brain[8–10,15]−reliably emerge from learning to multi-task in multi-dimensional environments. Together, our results indicate that abstract representations in the brain may be a consequence of – as well as a boon to[36]−complex behavior.

## Results
### Abstract representations allow knowledge to be generalized across contexts

The knowledge of latent structure that is present in the sensory world can enable generalization. For example, the appearance of different kinds of berries can be described by two continuous latent variables: color and shape. As an example, berries that have a similar shape are likely to also have similar texture when eaten, regardless of their color (Fig. 1a, left); further, berries that are red may taste more similar to each other, despite differences in shape, than they do to berries that

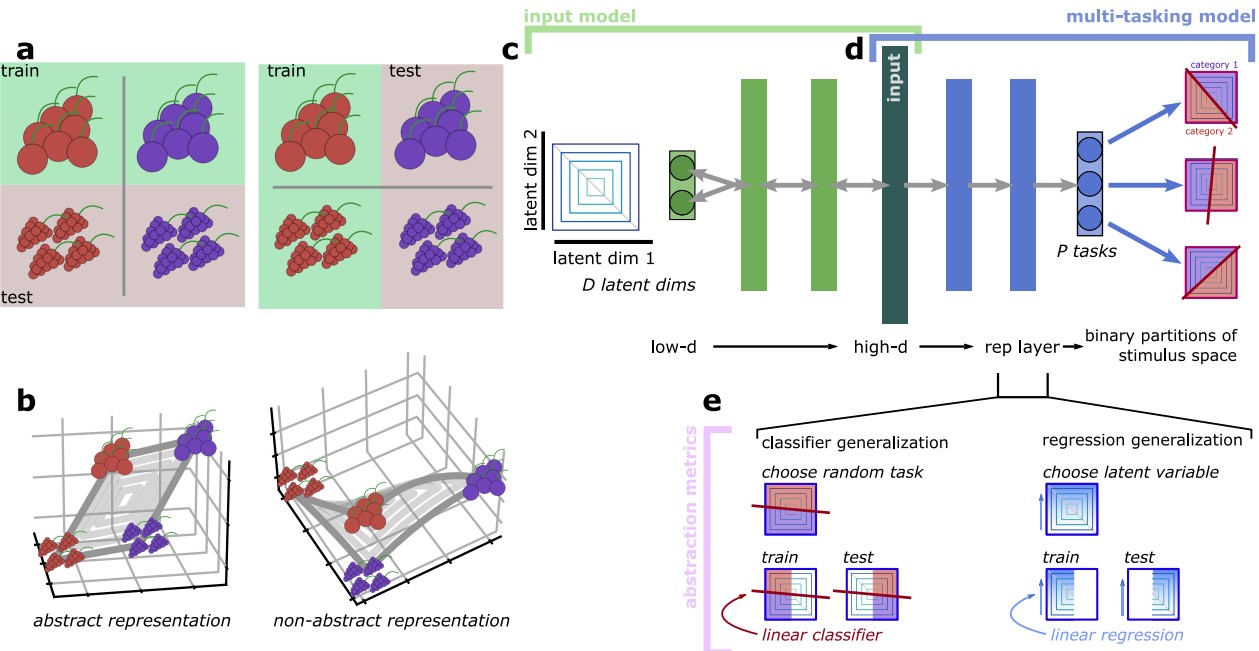

**Fig. 1 | The abstraction metrics and input representations. a** Two example classification tasks. (left) A classification learned between red and blue berries of one shape should generalize to other shapes. (right) A classification between red berries of two different shapes should generalize to blue berries of different shapes. **b** Examples of linear, abstract (left), and nonlinear, non-abstract (right) representations of the four example berries. **c** Schematic of the input model. **d** Schematic of the multi-tasking model. **e** Schematic of our two abstraction metrics, the classifier generalization metric (left) and the regression generalization metric (right).

are blue (Fig. 1a, right). Learning and taking advantage of this structure in the sensory world is important for animals that need to quickly react to novel stimuli using information from previously experienced stimuli.

We refer to neural representations that reflect this latent structure as abstract. In the example above, one form of an abstract representation of these latent variables is a linear representation of them in neural population activity, such a representation would have a low-dimensional, rectangular structure in neural population space (Fig. 1b, left); a non-abstract representation of these latent variables would have a higher-dimensional distorted structure, such as one created by neurons that each respond only to particular conjunctions of color and shape (Fig. 1b, right). The abstract representation has the desirable quality that, if we learned a neural readout that classifies blue berries from red berries using berries with only one shape (e.g., the two bottom berries in Fig. 1b, left), then we would not need to modify this classifier to apply it to berries of a different shape (e.g., the two top berries in Fig. 1b, left); while the same classification can be learned for the non-abstract representation, it will not generalize (compare the two berries to the left and to the right in Fig. 1b, right).

Here, we study how abstract representations like the ones in our example emerge for stimuli described by $D$ continuous latent variables in a feedforward neural network. The latent variables themselves are already abstract. So, we begin by constructing a nonlinear and non-abstract representation of the latent variables to use as our input going forward (Fig. 1c), which we refer to as the standard input. Then, we introduce the multi-tasking model, which receives these non-abstract representations of the latent variables as input (Fig. 1d, left) and is then trained to perform $P$ random binary classification tasks on the latent variables (Fig. 1d, right). Finally, after the multi-tasking model is fully trained, we quantify the level of abstraction developed in its representation layer using two abstraction metrics (Fig. 1e).

The first abstraction metric is referred to as the classifier generalization metric, and is nearly identical to the cross-category generalization performance used in previous work[8]. For the classifier generalization metric, we begin by selecting a novel categorization task on the latent variables (Fig. 1e, top left). Then, we train a linear classifier to perform that task using samples from the multi-tasking model representation layer that are taken from only one half of the latent variable space (Fig. 1e, bottom left, train). Then, we test this trained classifier on samples from the other half of the latent variable space (Fig. 1e, bottom left, test). If the classifier generalization performance is greater than chance, then this indicates that the representations developed in the multi-tasking model are at least partially abstract, because a category learned in one part of latent variable space successfully generalizes to another part of latent variable space. High classifier generalization performance has been observed for sensory and cognitive features in neural data recorded from the hippocampus and prefrontal cortex[8].

The second abstraction metric is referred to as the regression generalization metric (Fig. 1e, top right). This metric has the same structure as the classifier generalization metric, but uses a linear regression model instead of a linear classifier. Here, we begin by selecting a random latent variable. Then, we train a linear regression model to decode the value of that latent variable using samples from the multi-tasking model representation layer that are taken from only one-half of the latent variable space (Fig. 1e, bottom right, train). As before, we then test the trained linear regression model on samples from the other half of latent variable space (Fig. 1e, bottom right, test).

Metrics similar to both of these are often used in the machine learning literature[18,37]). The classifier generalization metric requires that the coarse structure of the representations be abstract, but is less sensitive to small deviations. The regression generalization metric is much stricter, and is sensitive to even small deviations from a representation that follows the underlying latent variable structure. In

some cases, we also compare these metrics of out-of-distribution generalization to standard cross-validated performance on the whole latent variable space. Intuitively, the standard cross-validated performance of both metrics serves as a best case for their out-of-distribution generalization performance (i.e., the case where what is learned from only half the representation space is just as informative about the global representation structure as what would be learned from the whole representation space). In a perfectly abstract representation, the standard and out-of-distribution generalization performances would be equal to each other.

Importantly, each of these three components of our framework is trained in sequence to each other: The input model (Fig. 1c) is trained first and then frozen. The input model is used to generate the training data for the multi-tasking model (Fig. 1d), which is trained second. Then, finally, we use our abstraction metrics (Fig. 1e) to quantify the level of abstraction present in the representation layer of the trained multi-tasking model (and in the trained standard input, as in Fig. 2).

### The input is sparse, high-dimensional, and non-abstract

First, we develop an input model to construct non-abstract representations of known $D$-dimensional latent variables, which we refer to as the standard input. In most of the paper, we assume $D = 5$ Gaussian-distributed latent variables, however, our results are similar for uniformly distributed latent variables (and see "A sensitivity analysis of the multi-tasking model and $\beta$VAE" in Supplementary Methods). The standard input is a feedforward autoencoder that receives the latent variables as inputs and is trained to satisfy two objectives: First, it must maximize the embedding dimensionality of activity in its representation layer (Fig. 2a, right, high-d input) and, second, to reconstruct the original stimulus using only the representation (Fig. 1a, blue arrows toward the left). That is, we want a high-dimensional representation of the latent variables that does not discard any information. The non-abstract representations generated by this procedure will be used as the input to the multi-tasking model.

After training, we visualize the response fields of units in the standard input representation layer (Fig. 2b). The response fields are sparse, conjunctive, and often multi-modal. We also compare the population representation of two latent variable dimensions prior to (Fig. 2c, left) and after Fig. 2c, right) undergoing this transformation. This visualization illustrates that the population representation also becomes highly disordered and tangled. In the full population, only approximately 4% of units are active for a given stimulus—and each individual unit is also highly sparse (Fig. 2d, left) according to a standard measure of sparseness (see "Quantifying sparseness and dimensionality" in Methods). Together, all of this leads to a large dimensionality expansion, from the $D = 5$-dimensional latent variables to a representation with an embedding dimensionality of close to 200, measured by the participation ratio[38] (and see "Quantifying sparseness and dimensionality" in Methods).

While high-embedding dimensionality and sparseness are already hallmarks of non-abstract representations, we also directly visualize, and then quantify, the level of abstraction in the standard input representations using the classification and regression generalization metrics that we developed for (and will later apply to) the representation layer of the multi-tasking model. We show that both a classifier (Fig. 2e, left) and a linear regression (Fig. 2e, right) trained only on one-half of the latent variable space (Fig. 2e, trained) achieve good performance in that region. However, in both cases, performance collapses when moving into the untrained region of latent variable space (Fig. 2e, tested). This is reflected in the full classification and regression generalization performance quantification: both classification and regression generalization performance is significantly decreased from generalization performance calculated on the latent variables themselves (Fig. 2f, green relative to blue dot) and relative to

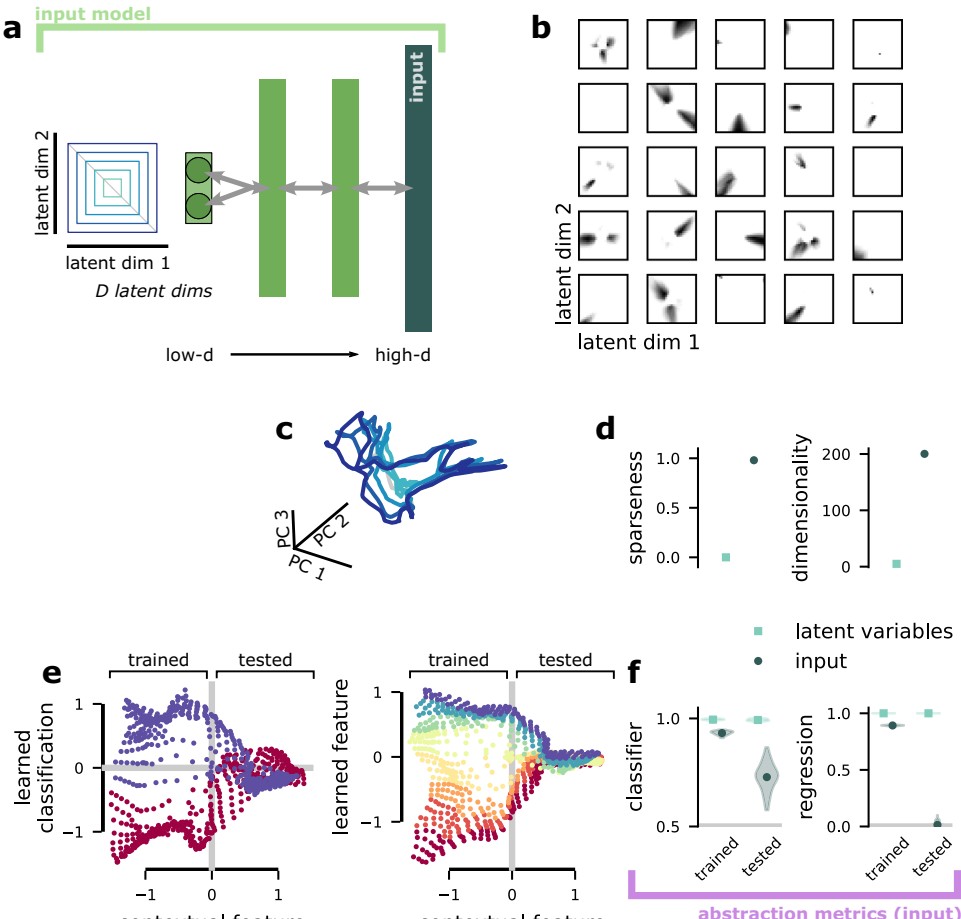

**Fig. 2 | The input model. a** Schematic of the input model. Here, schematized for $D = 2$ latent variables. The quantitative results are for $D = 5$. **b** The 2D response fields of 25 random units from the high-d input layer of the standard input. **c** The same concentric square structure shown to represent the latent variables in **a** after being transformed by the standard input. **d** (left) The per-unit sparseness (averaged across units) for the latent variables ($S = 0$, by definition) and standard input ($S = 0.97$). (right) The embedding dimensionality, as measured by participation ratio, of the latent variables (5, by definition) and the standard input (~190). **e** Visualization of the level of abstraction present in the high-d input layer of the standard input, as measured by the classifier generalization metric (left) and regression generalization metric (right). In both cases, the y-axis shows the distance from the learned classification hyperplane (right: regression output) for a classifier (right: regression model) trained to decode the sign of the latent variable on the y-axis (right: the value of the latent variable on the y-axis) only on representations from the left part of the x-axis (trained). The position of each point on the x-axis is the output of a linear regression for a second latent variable (trained on the whole latent variable space). The points are colored according to their true category (value). **f** The performance of a classifier (left) and regression (right) when it is trained and tested on the same region of latent variable space (trained) or trained in one region and tested in a non-overlapping region (tested, similar to **e**). Both models are trained directly on the latent variables and on the input representations produced by the standard input. The gray line is chance. The standard input produces representations with significantly decreased generalization performance.

the classification and regression performance in the trained region (Fig. 2f, right relative to left).

## The multi-tasking model learns abstract representations

To recover the abstract structure of the latent variables from the non-abstract representations produced by the standard input, we introduce the multi-tasking model (Figs. 1d and 3a). The multitasking model is a multilayer feedforward neural network model that is trained to perform $P$ different binary classification tasks (see "The multi-tasking model" in Methods for details). These tasks are analogous to the tasks that animals perform in many experimental settings, as described above. For instance, if an animal eats a berry, the animal later receives information about whether that berry was edible or poisonous. If we assume that the edibility of a berry is represented by one of our $D$ latent variables, then, in the multi-tasking model, this classification task corresponds to the model being trained to produce one output when the latent variable is positive and another output when the latent variable is negative. In the full model, the category boundary for each classification task is chosen to be a random hyperplane in the full $D$-

dimensional latent variable space (i.e., each task depends on multiple latent variables). In all of our analyzes, we focus on the representations of the stimuli that are developed in the layer preceding the task output layer, which we refer to as the representation layer (but see fig. S12 and "Abstraction emerges even in earlier layers of the multi-tasking model" in Supplementary Methods for an analysis of the other layers).

We show that the multi-tasking model develops fully abstract representations of the $D$ latent variables when trained to perform $P \geq D$ classification tasks. First, we visualize how the representations developed by our model compare to the abstract latent variables. In particular, we visualize the representations in the same three ways as we did the standard input (Fig. 2c, e, f). First, we compare a concentric square representation of the latent variables (Fig. 3a, left) to the same structure in the representation layer (Fig. 3b). For only a single task, the representations in the model collapse along a single dimension, which corresponds to the performance of that task (Fig. 3b, top). While this representation is not abstract, it does mirror distortions in sensory representations that are often observed when animals are overtrained on single tasks[39,40]. However, when we include a second task in the

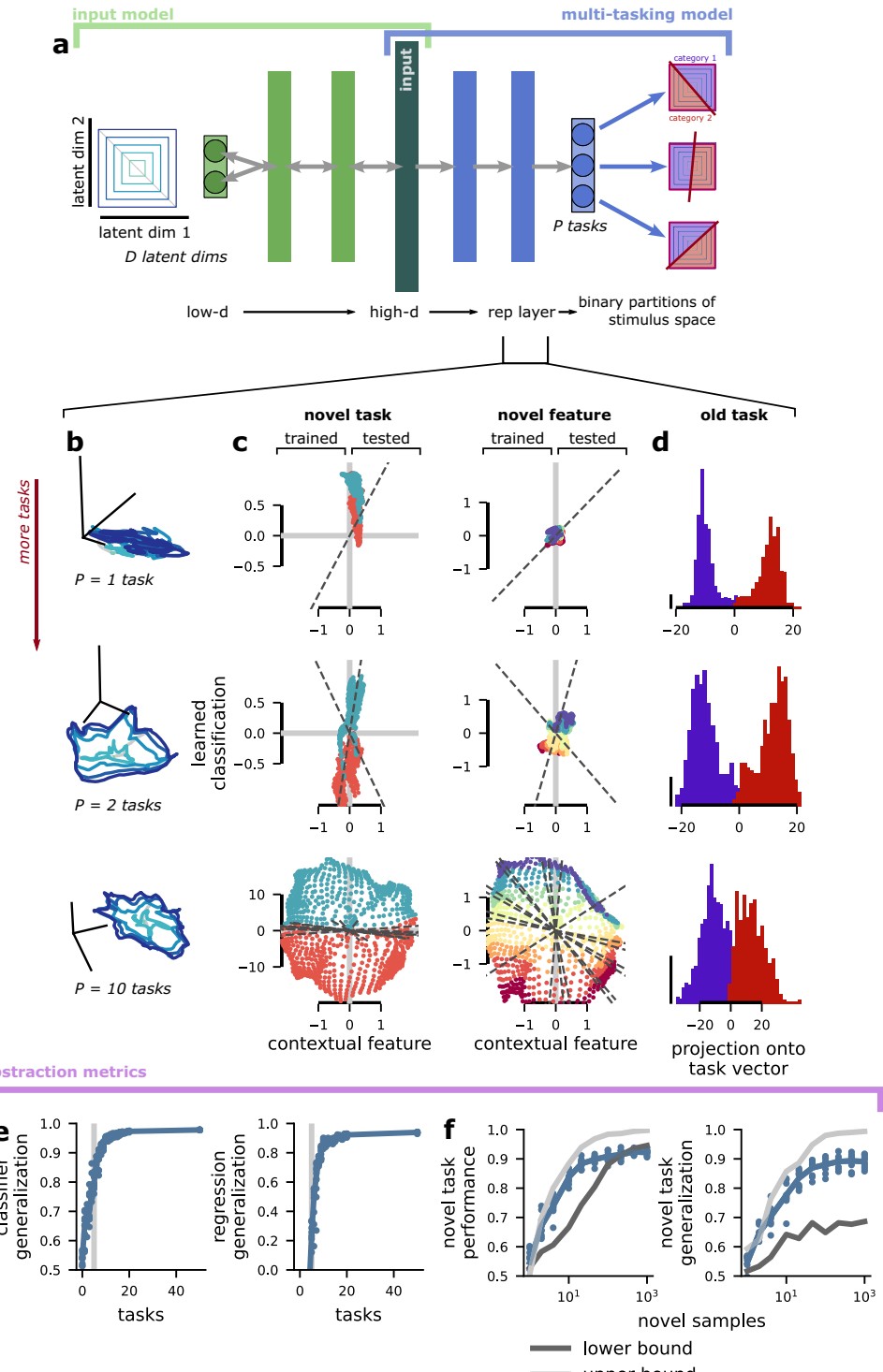

training procedure, abstract representations begin to emerge (Fig. 3b, middle). In particular, the representation layer is dominated by a two-dimensional abstract representation of a linear combination of two of the latent variables. Next, we demonstrate that this abstract structure becomes more complete as the number of tasks included in the training is increased. For $P = 10$ and $D = 5$, the visualization suggests that the representation has become fairly abstract (Fig. 3b, bottom).

We also visualized these results more directly using an approach similar to the classification and regression generalization metrics (and used above for the standard input, Fig. 2e). For each of the models, we train a linear classifier (regression) to decode the sign (the value) of one latent variable using samples drawn from only one half of latent

variable space (Fig. 3c, classification is left, regression is right, trained). Then, we visualize the output of that learned decoder as we move into the held out half of latent variable space (Fig. 3c, tested and compare with Fig. 2e). For one and two trained tasks (Fig. 3c, top and middle), the learned decoder performs poorly in both the trained and tested regions, because the network is highly specialized for the one (or two) tasks that it was trained to perform. However, for $P = 10$ trained tasks (Fig. 3c, bottom), the learned decoder performs well in both regions, indicating the emergence of fully abstract representations. Finally, we also visualize the projection of the representation layer from each of multi-tasking models onto one of the task outputs (Fig. 3d). We see that for one and two tasks (Fig. 3d, top and middle), the task output

**Fig. 3 | The emergence of abstraction from classification task learning.**
**a** Schematic of the multi-tasking model. It receives a high-dimensional input representation of $D$ latent variables (here, from the standard input, as shown in Fig. 1e, left) and learns to perform $P$ binary classifications on the latent variables. We study the representations that this induces in the layer prior to the output: the representation layer. **b** Visualization of the concentric square structure as transformed in the representation layer of a multi-tasking model trained to perform one (top), two (middle), and ten (bottom) tasks. The visualization procedure is the same as Fig. 2c. **c** The same as **b**, but for visualizations based on classifier (left) and regression (right) generalization. The classifier (regression) model is learned on the left side of the plot, and generalized to the right side of the plot. The output of the model is given on the $y$ axis and each point is colored according to the true latent variable category (i.e., sign) or value. The visualization procedure is the same as Fig. 2e. The visualization shows that generalization performance increases with the number of tasks $P$ (increasing from top to bottom). **d** The activation along the output dimension for a single task learned by the multi-tasking model for the two different output categories (purple and red). The distribution of activity is bimodal for multi-tasking models trained with one or two tasks, but becomes less so for more tasks. **e** The classifier (left) and regression (right) metrics applied to model representations with different numbers of tasks. **f** The standard (left) and generalization (right) performance of a classifier trained to perform a novel task with limited samples using the representations from a multi-tasking model trained with $P = 10$ tasks as input. The lower (dark gray) and upper (light gray) bounds are the standard or generalization performance of a classifier trained on the input representations (lower) and directly on the latent variables (upper). Note that the multi-tasking model performance is close to that of training directly on the latent variables in all cases.

value is strongly separated and bimodal. These representations suggest that the multi-tasking model is discarding most information about the latent variables except that which is necessary to solve the tasks— and also illustrates why we would have poor performance when attempting to learn a novel task using the representation (Fig. 3c, top and middle). However, for $P = 10$ tasks, the task outputs are less separated and appear more continuous. This suggests that the multi-tasking model develops more information about the latent variables that could underlie both novel task learning and abstract representations (Fig. 3c, bottom).

Next, we quantify how the level of abstraction developed in the representation layer depends on the number of classification tasks used to train the model (Fig. 3c). For each number of classification tasks, we train 10 multi-tasking models to characterize how the metrics depend on random initial conditions. As the number of classification tasks $P$ exceeds the number of latent variables $D$, both the classification and regression generalization metrics saturate to near their maximum possible values (classifier generalization metric: exceeds 90% correct with 8 tasks; regression generalization metric: exceeds $r^2 = 0.8$ with 9 tasks; Fig. 3c, right of the gray line). Saturation of the classifier generalization metric indicates that the broad organization of the latent variables is perfectly preserved; while saturation of the regression generalization metric indicates that even the magnitude information that the multi-tasking model did not receive supervised information about is preserved and represented in a fully abstract format. Importantly, both the training and testing set split and the classification boundary for the classifier generalization metric are randomly selected —they are not the same as classification tasks used in training.

The multi-tasking model also reduces the number of samples required to both learn and generalize novel tasks. For a multi-tasking model trained to perform $P = 10$ tasks with $D = 5$ latent variables, we show how the performance of a novel classification task depends on the number of samples. We compare this performance to a lower bound (Fig. 3f, dark gray), from when the task is learned from the standard input representation; as well as an upper bound (Fig. 3f, light gray), from when the task is learned directly from the latent variables. The performance of the multi-tasking model nearly saturates this upper bound (Fig. 3f, left). Next, we perform a similar novel task analysis, but where the novel task is learned from one half of the stimulus space and is tested in the other half – just like our generalization analysis above (and schematized in Fig. 1c, top). We compare the same lower and upper bound as before and show that, again, the multi-tasking model representation nearly saturates the upper bound (Fig. 3d, right). Thus, not only does the multi-tasking model produce representations with good generalization properties, it also produces representations that lend themselves to the rapid (i.e., few sample) learning of novel tasks.

Next, we test how robust these abstract representations are to increases in the embedding dimensionality of the input, to changes to the classification tasks themselves, and to a different input type. First,

we show that this finding is almost unchanged given standard input models that produce higher-dimensional input (fig. S10 and see "The effect of increased input dimensionality on abstraction" in Supplementary Methods). Then, we show that our finding holds for three manipulations to the task structure. First, we show that unbalanced tasks (e.g., a more or less stringent criteria for judging the ripeness of a fruit—so either many more of the fruit are considered ripe than spoilt or vice versa; fig. S2a, top left; see "Unbalanced task partitions" in Methods for more details) have a negligible effect on the emergence of abstract representations (classifier generalization metric: exceeds 90% correct with 9 tasks, regression generalization metric: exceeds $r^2 = 0.8$ with 9 tasks; fig. S2b). Second, we show that contextual tasks (e.g., determining the ripeness of different fruits that occupy only a fraction of latent variable space; fig. S2a, top right; see "Contextual task partitions" in Methods for more details) produce only a moderate increase in the number of tasks required to learn abstract representations (classifier generalization metric: exceeds 90% correct with 14 tasks, regression generalization metric: exceeds $r^2 = 0.8$ with 14 tasks; fig. S2b). Third, we show that using training examples with information from only a single task (e.g., getting only a single data point on each trip to the store; fig. S2a, bottom, see "Partial information task partitions" in Methods for more details) also only moderately increase the number of tasks necessary to produce abstract representations (classifier generalization metric: exceeds 90% correct with 11 tasks, regression generalization metric: exceeds $r^2 = 0.8$ with 14 tasks; fig. S2b).

Thus, the multi-tasking model reliably produces abstract representations even given substantial heterogeneity in the amount of information per stimulus example and the form of that information relative to the latent variables. In the case of contextual tasks, the latent variable information provided by the tasks is necessarily partial. To develop abstract representation even in this case, the multi-tasking model must combine information from multiple different contextual tasks. Further, these results are also robust to variation in architecture: Changing the width, depth, and several other parameters of the multi-tasking model have only minor effects on classification and regression generalization performance (fig. S7 and see "A sensitivity analysis of the multi-tasking model and $\beta$VAE" in Supplementary Methods). The result is also robust to $L_1$ and $L_2$ regularization of the activity in the representation layer, which also increases the sparseness of that activity (fig. S9 and see "The effect of activity regularization on abstraction" in Supplementary Methods for more detail).

Finally, we ask whether the multi-tasking model can produce abstract representations from a different kind of input, chosen to mimic the structure of a population of highly local Gaussian receptive fields (RF), which are thought to be used to encode many kinds of stimuli across early sensory systems[24–28]. Here, we construct a representation of a $D = 5$ latent variables using randomly positioned Gaussian receptive fields (fig. S4a, left, and see "Abstract structure can be learned from early sensory-like representations" in Supplementary

Methods). These inputs have a highly curved geometry in population space (fig. S4a, right). Then, we show that the multi-tasking model recovers fully abstract representations from this highly local input (fig. S4c). Later, we explore two additional input types.

## Understanding the learning dynamics that produce abstract representations

The multi-tasking model is trained to simultaneously produce output for $P$ different random tasks. Importantly, the standard input used in this section already has high classification performance for random hyperplane tasks on the latent variables (Fig. 2f, left), due to its high-embedding dimensionality[4]. So, one possibility is that the representation layer in the multi-tasking model would retain the same, non-abstract structure. However, our results in the previous section and experiments with multi-tasking models that are trained with layers that all have the same width as the input (see "The effect of constant layer widths on abstraction" in Supplementary Methods and fig. S11) show that this is not the case. Instead, the multi-tasking model develops robustly abstract representations (Fig. 3e, f).

To understand why this occurs, we show that the training process increases the strength of the representation of an approximately $\min(P,D)$-dimensional component of the activity in the representation layer of the multi-tasking model. In particular, for a simplified multi-tasking model with a linear output layer, the loss for a particular sample $\mathbf{x}$ has the form,

$$L(\mathbf{x}) = \frac{1}{2}\sum_{i}^{P}[\text{sign}(A\mathbf{x}) - Wr(\mathbf{x})]_i^2 \qquad (1)$$

where $W$ are the weights connecting the representation layer to the task outputs, $r(\mathbf{x})$ is the activity corresponding to stimulus $\mathbf{x}$ in the representation layer, and $A$ is a $P \times D$ matrix of randomly selected task vectors (i.e., the vectors that define the binary classification hyperplane). This loss is minimized by making $r(\mathbf{x})$ a linear transform of $\text{sign}(A\mathbf{x})$. In backpropagation, this is achieved by increasing the strength of a component of $r(\mathbf{x})$ that has the same dimensionality as $\text{sign}(A\mathbf{x})$ (see "The dimensionality of representations in the multi-tasking model" in Methods). So, we show that,

$$\dim(\mathbb{E}_X \text{sign}(A\mathbf{x})\,\text{sign}(A\mathbf{x})^T) \approx \dim(\mathbb{E}_X A\mathbf{x}\mathbf{x}^T A^T) \qquad (2)$$

$$= \min(P, D) \qquad (3)$$

and the approximation becomes closer as $D$ becomes larger (and, indeed, we see less abstract representations for lower $D$, see "The dependence of learned abstract representations on latent variable dimensionality" in Supplementary Methods for more discussion). This means that, given application of backpropagation, the representation layer will tend to be dominated by a $\min(P,D)$-dimensional representation of the latent variables. Since this representation must also be able to satisfy the $P$ tasks, it will at least have high classifier generalization performance and may even have high regression generalization performance (see "Four possibilities for representations in the multi-tasking model" in Supplementary Methods for more discussion of alternative representations). While the multi-tasking model used in the rest of the paper has a sigmoid output nonlinearity, the intuition developed in this simplified case still applies.

## Abstract representations only emerge when task-relevant

Abstract representations for the latent variables do not emerge when the multi-tasking model is trained to perform random, highly nonlinear tasks. This follows what would be expected in the natural world: latent variables are learned as a way to solve multiple related tasks and to generalize knowledge from one task to another, rather than for their

own sake. Then, we show that abstract representations are recovered when the multi-tasking model learns a combination of latent variable-aligned and unaligned tasks.

First, we construct grid classification tasks, in which the latent variable space is divided into grid chambers, where each chamber has a roughly equal probability of being sampled (Fig. 4a, red lines). Then, we randomly assign each of the grid chambers to one of two categories (Fig. 4a, coloring; see "Grid classification tasks" in Methods for more details). In this case, there is nothing in the design of the multi-tasking model that privileges a representation of the original latent variables, since they are no longer useful for learning to perform the multiple grid classification tasks. Consequently, the multi-tasking model does not recover a representation of the original latent variables (Fig. 4b, c).

To make this intuition about the grid tasks more explicit, we show that—in contrast to the latent variable-aligned tasks that we have been using so far—the outcomes from a particular grid task are likely to be only weakly correlated with the outcomes from a different, randomly chosen grid task (Fig. 4d). Thus, rather than having a $D$-dimensional structure for $P \gg D$ tasks, the grid tasks will have a roughly $P$-dimensional structure for $P$ tasks. As expected, the multi-tasking model fails to learn a strongly abstract representation of the original latent variables, and the representation becomes less abstract as the grid tasks become higher-dimensional (i.e., when the grid has more chambers; Fig. 4c, middle and right, blue and purple lines).

Next, we examine the representations learned by the multi-tasking model when it must perform a mixture of latent variable-aligned and grid classification tasks (Fig. 4e). This situation is also chosen to mimic the natural world, as a set of latent variables may be relevant to some behaviors (the latent variable-aligned classification tasks), but an animal may need to perform additional behaviors on the same set of stimuli that do not follow the latent variable structure (the grid classification tasks, Fig. 4e, right). Here, we train the multi-tasking model to perform a fixed number of latent variable-aligned tasks, which are sufficient to develop an abstract structure in isolation (here, 15 tasks). However, at the same time, the model is also being trained to perform various numbers of grid tasks (Fig. 4f, x axis). While increasing the number of grid tasks does moderately decrease the abstractness of the developed representation (visualization: Fig. 4f; quantification: Fig. 4g), the multi-tasking model retains strongly abstract representations even while performing more than 45 grid tasks—3 times as many as the number of latent variable-aligned tasks.

Intuitively, this occurs because the latent variable-aligned tasks are correlated with each other and follow the structure of the $D$-dimensional latent variable space, while each of the grid tasks has low correlation with any other grid task (Fig. 4d). Thus, a shared representation structure is developed to solve all the latent variable-aligned tasks essentially at once, while a smaller nonlinear component is added on to solve each of the grid tasks relatively independently. Interestingly, the combination of abstract structure with nonlinear distortion developed by the multi-tasking model here has also been observed in the brain and other kinds of feedforward neural networks (though learning tasks analogous to our grid tasks was not necessary for it to emerge)[8]. We believe that this compromise between strict abstractness (which allows for generalization) and nonlinear distortion (which allows for flexible learning of random tasks[4,5]) is fundamental to the neural code.

## The multi-tasking model learns abstract representations from other kinds of nonlinear inputs

To understand the constraints on the multi-tasking model's ability to learn abstract representations from non-abstract input, we introduce both a new input model (Fig. 5a–c) and a new kind of nonlinear task (Fig. 5a, d, e). We control the length scale of correlations in both the input model and the tasks. Then, we quantify the classification and

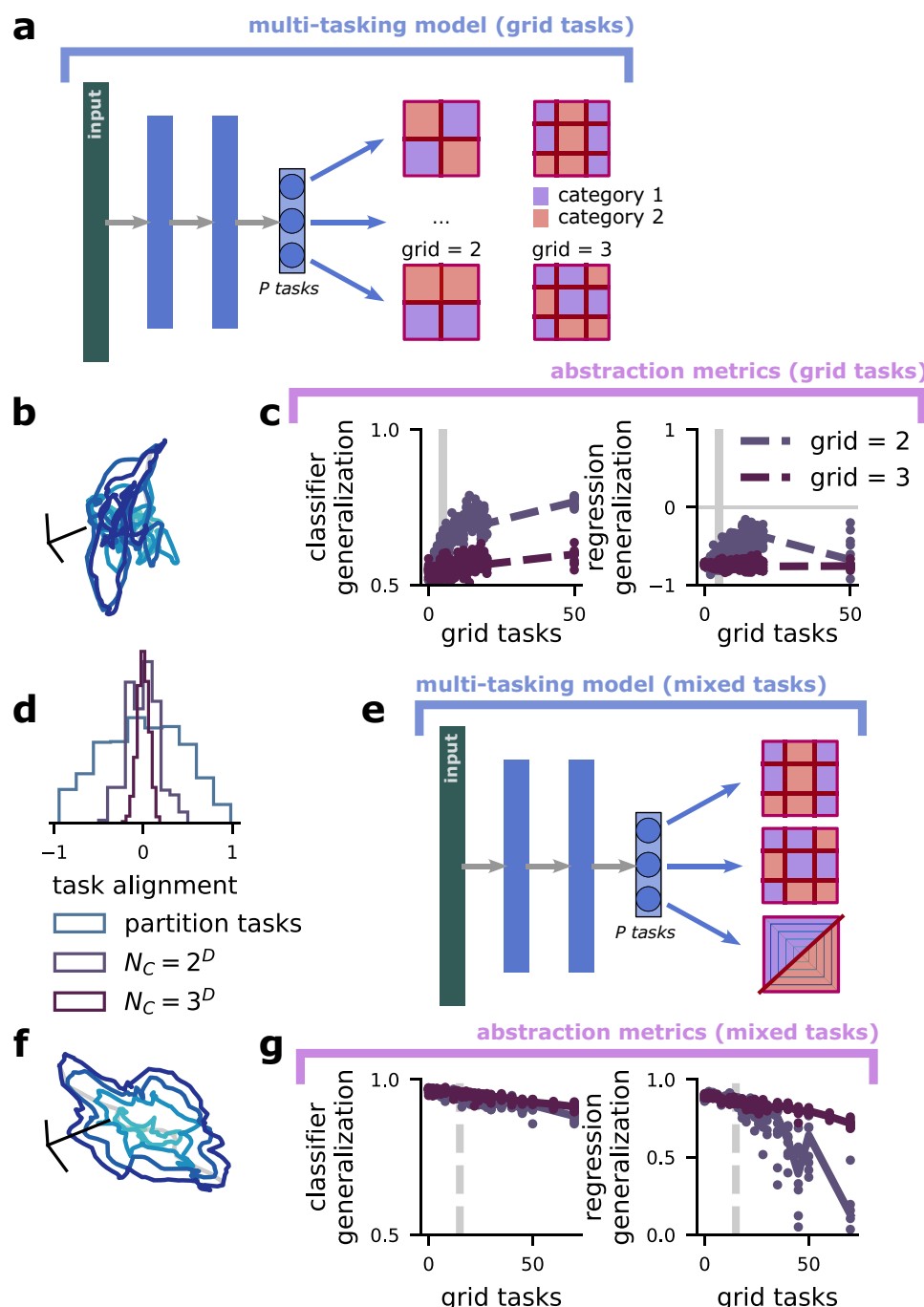

**Fig. 4 | Abstract representations emerge for heterogeneous tasks, and in spite of high-dimensional grid tasks. a** Schematic of the multi-tasking model with grid tasks. They are defined by the grid size, grid, the number of regions along each dimension (top: grid = 2; bottom: grid = 3), and the number of latent variables, $D$. There are grid$^D$ total grid chambers, which are randomly assigned to category 1 (red) or category 2 (blue). Some grid tasks are aligned with the latent variables by chance (as in top left), but this fraction is small for even moderate $D$. **b** Visualization of the representation layer of a multi-tasking model trained only on grid tasks, with $P = 15$. **c** Quantification of the abstraction developed by a grid task multi-tasking model. (left) Classifier generalization performance. (right) Regression generalization performance. **d** The alignment (cosine similarity) between between

randomly chosen tasks for latent variable-aligned classification tasks, $n = 2$ and $D = 5$ grid tasks, and $n = 3$ and $D = 5$ grid tasks. **e** Schematic of the multi-tasking model with a mixture of grid and linear tasks. **f** Same as **b**, but for a multi-tasking model trained with a mixture of: $P = 15$ latent variable-aligned classification tasks and a variable number of grid tasks ($x$ axis). **g** Same as **c**, but for a multi-tasking model trained with $P = 15$ latent variable-aligned classification tasks and a variable number of grid tasks. While the multi-tasking model trained only with grid tasks does not develop abstract representations, the multi-tasking model trained with a combination of grid and linear tasks does – even when the number of grid tasks outnumbers the number of linear tasks.

regression generalization performance as we vary both length scales simultaneously (Fig. 5f, g).

For the input model, we use random Gaussian processes with radial basis function kernels of different length scales. To illustrate this approach, we begin with a $D = 1$ normally distributed latent variable

(Fig. 5a, left), then generate random Gaussian process functions that map this variable to a scalar output (three functions in Fig. 5a, center). The scalar outputs of many random Gaussian process functions are then used as input to the multi-tasking model (Fig. 5a, right; and see "Random Gaussian process inputs" in Methods for more details).

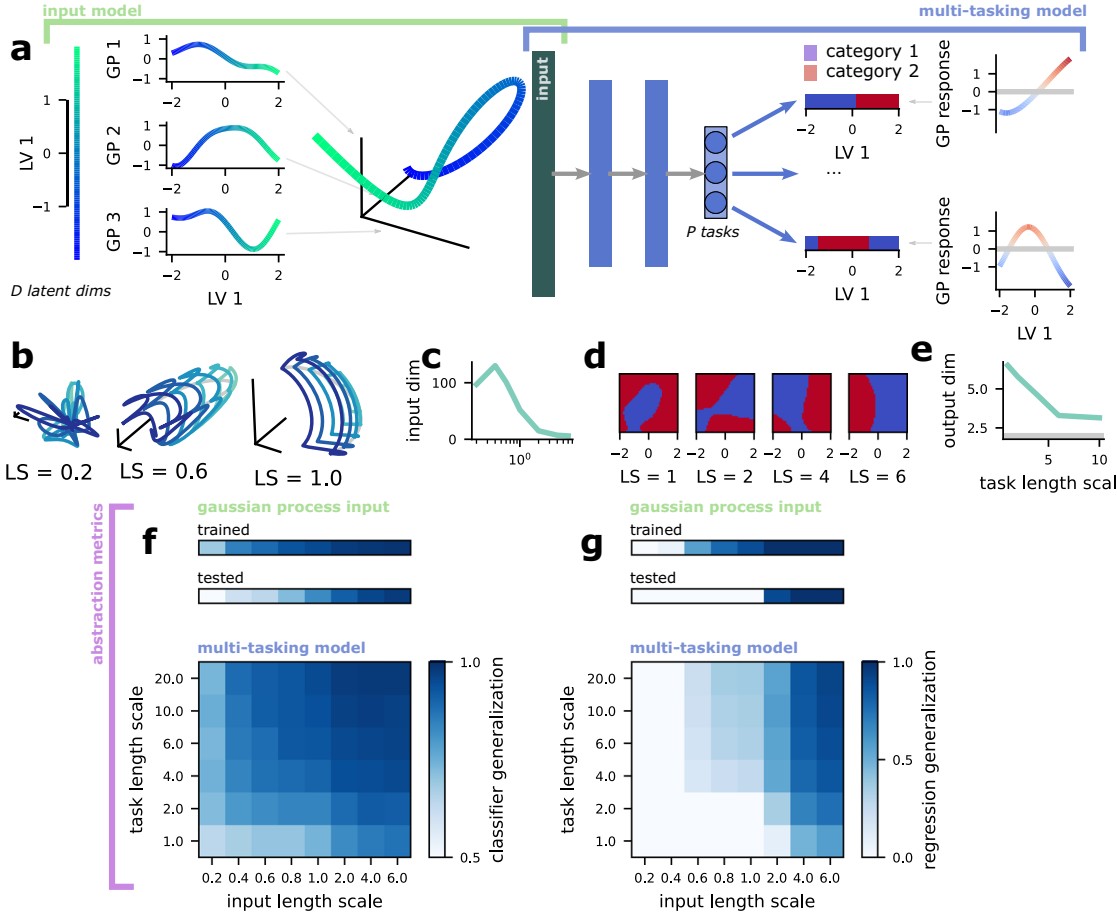

**Fig. 5 | The multi-tasking model learns abstract structure from both random Gaussian process inputs and output tasks. a** (left) Schematic of the creation of a random Gaussian process input for one latent variable dimension ($D = 1$). Random Gaussian processes with a length scale = 1 are learned for the single latent variable shown on the left. Then, the responses produced by these random Gaussian processes are used as input to the multi-tasking model. The full random Gaussian process input has 500 random Gaussian process dimensions and $D = 5$ latent variables. (right) Schematic of the creation of two random Gaussian process tasks for the $D = 1$-dimensional latent variable shown on the left, showing both two example binary classification tasks and the random Gaussian process that is thresholded at zero to create each task. **b** Visualization of the input structure for random Gaussian process inputs of different length scales. **c** The embedding dimensionality (participation ratio) of the random Gaussian process for different length scales. Note that it is always less than the dimensionality of 200 achieved by the standard input. **d** Examples of random Gaussian process tasks for a variety of length scales. The multi-tasking model is trained to perform these tasks, as schematized in **a**. **e** The embedding dimensionality (participation ratio) of the binary output patterns required by task collections of different length scales. **f** Classifier generalization performance of a multi-tasking model trained to perform $P = 15$ classification tasks with $D = 5$-dimensional latent variables, shown for different conjunctions of task length scale (changing along the $y$ axis) and input length scale (changing along the $x$ axis). **g** Regression generalization performance shown as in **f**.

Where we show results from these random Gaussian process inputs, we use $D = 5$ rather than the $D = 1$ used in the example.

The length scale of the random Gaussian process kernel controls how far two points need to be from each other in latent variable space before they become uncorrelated in representation space. As a result, the length also controls how nonlinear and non-abstract the resulting input representation of the latent variables is (Fig. 5b). In particular, a low length scale (e.g., <1) means that the input representation is both relatively high-dimensional (Fig. 5c, left) and non-abstract (Fig. 5f, g, gaussian process input). Alternatively, a high length scale (e.g., >4) produces low-dimensional (Fig. 5c, right) and abstract representations (Fig. 5f, g, gaussian process input). We show that the multi-tasking model achieves high classifier generalization performance for all random Gaussian process input length scales that we investigated (Fig. 5f, multi-tasking model, top row). We also show that the multi-tasking model achieves moderate regression generalization performance for many different length scales as well, though regression generalization performance remains at chance for the shortest length scales that we investigated (Fig. 5g, multi-tasking model, top row).

The random Gaussian process input differs from our previous input type in that, for low-length scales, a linear decoder cannot

reliably learn random categorical partitions (as is the case for the standard input, see Fig. 1f). The random Gaussian process representations also have significantly lower participation ratios than those produced by the standard input. We can see that the random Gaussian process input tends to fold back on itself for low length scales (Fig. 5a). This increased folding may explain the lower embedding dimensionality of the random Gaussian process relative to the standard input; we also believe that it would increase the complexity of the transformation required to produce abstract representations, which may explain the lower regression generalization performance for the random Gaussian process inputs.

## The multi-tasking model learns abstract structure from tasks with nonlinear curvature

While we have shown that the multi-tasking model learns abstract structure from several different manipulations of linear tasks (fig. S2a, b) and fails to learn abstract structure from highly nonlinear tasks, for which the latent variables themselves are no longer relevant (Fig. 4a, b), these two examples represent relatively extreme cases. Here, we show that the multi-tasking model still produces abstract representations in many cases in between

these two extremes, when it is trained on tasks with different levels of nonlinear curvature (Fig. 5f, g). To produce these tasks, we generate random Gaussian processes with radial basis function kernels of a particular length scale (Fig. 5a, right), then use them to produce two distinct categories by binarizing their output (where outputs >0 are in one category and ≤0 are the other; Fig. 5a, right, and see "Random Gaussian process tasks" in Methods for more details).

Following this procedure, we produce tasks with a variety of length scales (Fig. 5d, length scale increases from left to right). Similar to the random Gaussian process input, tasks with lower length scales will have more curved boundaries and multiple distinct category regions, similar to the grid tasks (Fig. 5d, left); tasks with higher length scales will tend to have less curved boundaries—and large length scales (e.g., >10) will approximate the linear tasks from before. We quantify the nonlinearity of these tasks by computing how the embedding dimensionality of the required output depends on task length scale for $P = 15$ classification tasks on $D = 2$ latent variables (Fig. 5e). As discussed above, the nonlinear grid tasks have an output dimensionality that approaches the number of tasks, while the linear tasks used above have an output dimensionality that is only slightly higher than the number of latent variables. We show that the random Gaussian process tasks have a required output dimensionality that lies between these extremes, and that decreases with increased length scale (Fig. 5e). Our theory suggests that the multi-tasking model will learn abstract representations for moderate levels of curvature.

We show that a multi-tasking model trained to perform $P = 15$ classification tasks produces representations with above-chance classifier generalization performance for all task length scales that we investigated (Fig. 5f, multi-tasking model, middle columns). Further, the multi-tasking model produces representations with above-chance regression generalization performance for many different task length scales as well (Fig. 5g, multi-tasking model, middle columns), though it is less consistent than classifier generalization performance. Thus, the multi-tasking model produces partially abstract representations even from highly curved and sometimes multi-region task boundaries, and produces fully abstract representations for curved task boundaries. So, while the multi-tasking model does not produce abstract representations in the extreme case of the highly nonlinear grid tasks, it does produce abstract representations for many intermediate task structures (shown both here and above).

Finally, instead of training the multi-tasking model using random Gaussian process tasks, we explored whether or not the network representations could be used to efficiently learn and generalize on novel random Gaussian process tasks instead of the linear tasks that we have been using to quantify abstraction so far. We found that, across several different length scales, both the sample efficiency and generalization performance on the novel, curved task were close that of learning directly from the latent variables (fig. S13 and see "Novel random Gaussian process task learning" in Supplementary Methods for more detail; this mirrors the efficiency and generalization performance of learning a novel linear classification task, Fig. 3f). Thus, the abstraction representations learned by the multi-tasking model facilitate efficient learning and generalization even when the novel task is not linear.

## The multi-tasking model produces abstract representations from image inputs

Given the previous results showing that the multi-tasking model produces only partially abstract representations from highly tangled inputs (i.e., the low length scale random Gaussian process inputs explored in Fig. 5), we next asked whether the multi-tasking model would produce fully (i.e., high classification and regression generalization performance) or partially (i.e., only high classifier generalization performance) abstract representations of the image inputs often used to study disentangling in the machine learning literature (e.g.,[41]): A chair image dataset that includes 3D rotations[42] and a simple 2D shape dataset[43] (Fig. 6a). First, we pre-process the images using a deep network trained to perform object recognition (see "Pre-processing using a pre-trained network" in Methods). These networks have been shown to develop representations that resemble those found in brain regions like the inferotemporal cortex (ITC)[44], at the apex of the primate ventral visual stream. Then, we use a two-layer network to learn several distinct classification tasks which partition the space of the latent variables that describe the images (Fig. 6b, the same kinds of tasks as in Fig. 3 and see "The image datasets" in Methods for more details).

The images in both datasets are described by three continuous parameters and one categorical variable. The chair images have continuous horizontal position, vertical position, and azimuthal rotation variables, along with the categorical chair type variable. The 2D shape images are described by continuous horizontal position, vertical position, and scale variables, along with the categorical shape type variable. For both datasets, the tasks learned by the model depend only on the continuous variables, not on the categorical variables.

In both datasets, the pixel-level images (Fig. 6c, e, top) and the representations produced by the pre-trained network alone (Fig. 6c, e, bottom) are non-abstract. However, the representations produced by the multi-tasking model are abstract, and show strong classifier generalization performance and moderately high regression generalization performance (Fig. 6d, f). Thus, the multi-tasking model can produce fully abstract representations from representations of objects similar to those observed in the brain.

We also explore several other kinds of generalization using these image inputs. First, we train the multi-tasking model using only a subset of the different chairs (shapes) and then perform the generalization analysis in the usual way, but using only the chairs (shapes) that were held out (fig. S8 and see Zero-shot categorical generalization for image inputs in Supplementary Methods for more detail). Here, we find fully abstract representations for the shape representations and partially abstract representations for the chairs (fig. S8b). Next, we perform a similar analysis, but train our abstraction metric models using chairs (shapes) that were used during multi-tasking model training and test them on the held out chairs (shapes; fig. S8c). Here, we find the same result as above: fully abstract representations for the shapes and partially abstract representations for the chairs (fig. S8d).

Finally, we also use the image setting to investigate one important property of abstract representations that is not captured by the standard multi-tasking model: compositionality of representations. In machine learning, abstract representations are desirable primarily because they allow representations to be composed to produce output representations with predictable features[41]. To investigate this in our setting, we train a multi-tasking model on the shape image dataset, where the multi-tasking model must perform binary tasks as before, but is also tasked with reconstructing the original image input from the representation layer as well (see "The multi-tasking model can be used as an abstract, generative model" in Supplementary Methods for the details of the model). Then, we learn a vector representation of shape scale from two of the three shapes included in the dataset (fig. S5d). Next, we take the representation for the third shape at a starting scale and use the learned vector to produce shape examples with increased and decreased scale (fig. S5e). Thus, not only does the representation of scale generalize across the different shapes, but this property can be used to generate images with a desired scale in a compositional way.

## The multi-tasking model learns abstract representations using reinforcement learning

In all of the previous cases, we have used supervised learning to train the multi-tasking model. While this is widely used in machine learning and has been shown to produce representations that resemble those found in the brain in many cases[44–46], the information used to train the network during supervised training is qualitatively different from

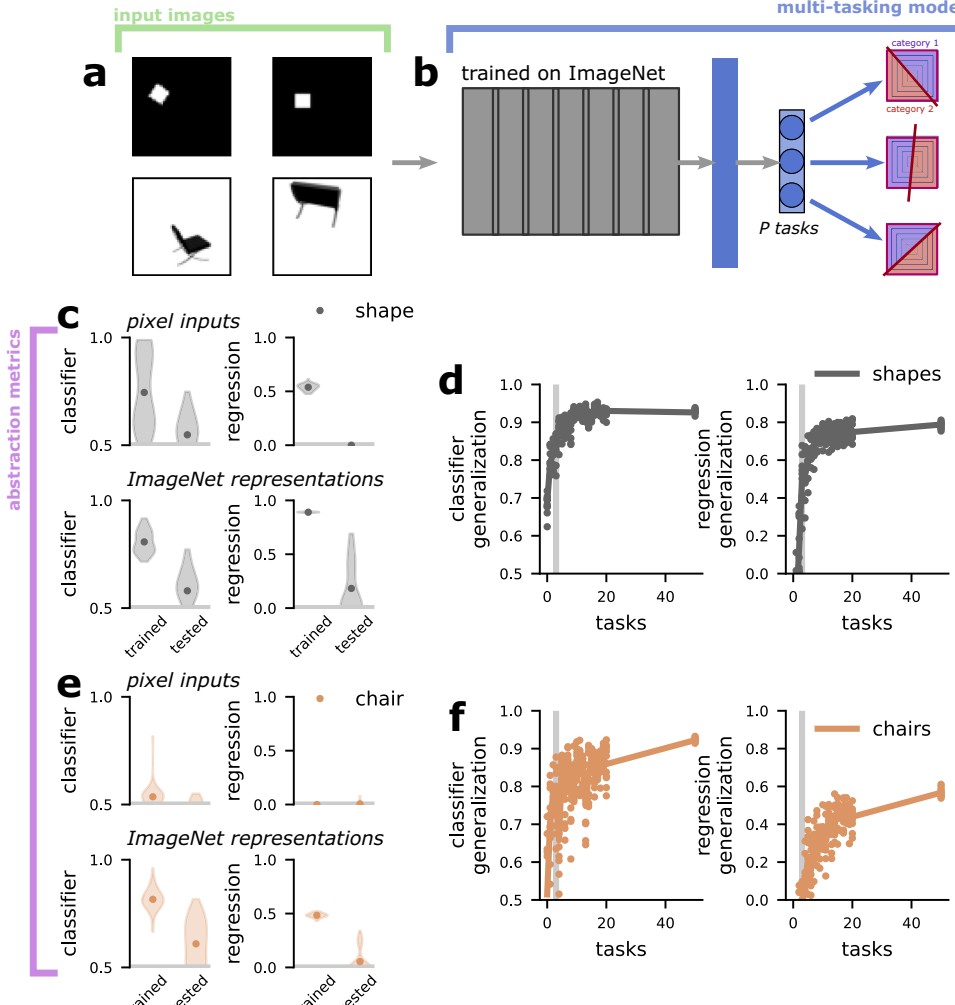

**Fig. 6 | The multi-tasking model produces abstract representations from image inputs. a** Examples from the 2D shape dataset[43] (top) and chair image dataset[42] (bottom). The 2D shapes dataset is from: Matthey, L., Higgins, I., Hassabis, D. & Lerchner, A. dsprites: Disentanglement testing sprites dataset. https://github.com/deepmind/dsprites-dataset/ (2017). **b** Schematic of modified model. The multi-tasking model now begins with a networked pre-trained on the ImageNet challenge, followed by a few additional layers of processing before learning binary tasks as before (see "Pre-processing using a pre-trained network" in Methods). **c** The classifier (left) and regression (right) generalization performance when applied to the shape image pixels (top) or ImageNet representations (bottom). **d** The classifier (left) and regression (right) generalization performance of the multi-tasking model on the shape images. **e** The same as **c** but for the chair images. **f** The same as **d** but for the chair images.

the information that would be received by a behaving organism performing multiple tasks. Here, we confirm that the multi-tasking model still produces abstract representations when trained using reinforcement learning.

We use a modified version of the deep deterministic policy gradient (DDPG)[47] reinforcement learning framework to train our networks. In this setup, there are two networks: an actor-network, which is trained to take a stimulus and produce the action (or set of actions) that will maximize reward (Fig. 7a, actor) – this is directly analogous to the full multi-tasking model as previously described. In reinforcement learning, the actor cannot be trained directly from the gradient between the produced and correct actions. So, instead, a second network, referred to as the critic, is created, which is trained to predict the reward outcome from an observation and a potential action (Fig. 7a, critics). The critic network is trained to accurately predict the reward that results from a stimulus-action pair. Then, the actor network is trained to produce actions that lead to predicted reward (see "The reinforcement learning multi-tasking model" in Methods for more details). Here, we create a critic network for each of the tasks that the reinforcement learning multi-tasking model is trained to perform.

While the reinforcement learning multi-tasking model learns the tasks less reliably than the supervised multi-tasking model (Fig. 7b, c), it still produces fully abstract representations for around ten trained tasks ($D = 5$, Fig. 7d). Interestingly, while some tasks are not successfully learned during the allotted training time, the learned tasks transition from near-chance performance to near-perfect performance within just a few training epochs (Fig. 7b). This suggests that additional hyperparameter tuning could potentially improve task learning consistency and push the representations to be even more strongly abstract with fewer trained tasks. However, such extensive tuning is out of the current scope of this work.

## Discussion

We demonstrate that requiring a feedforward neural network to perform multiple tasks reliably produces abstract representations. Our results center on artificial neural networks; however, we argue that abstract representations in biological neural systems could be produced through the same mechanism, as behaving organisms often need to multi-task in the same way as we have modeled here. We show that the learning of these abstract representations is remarkably

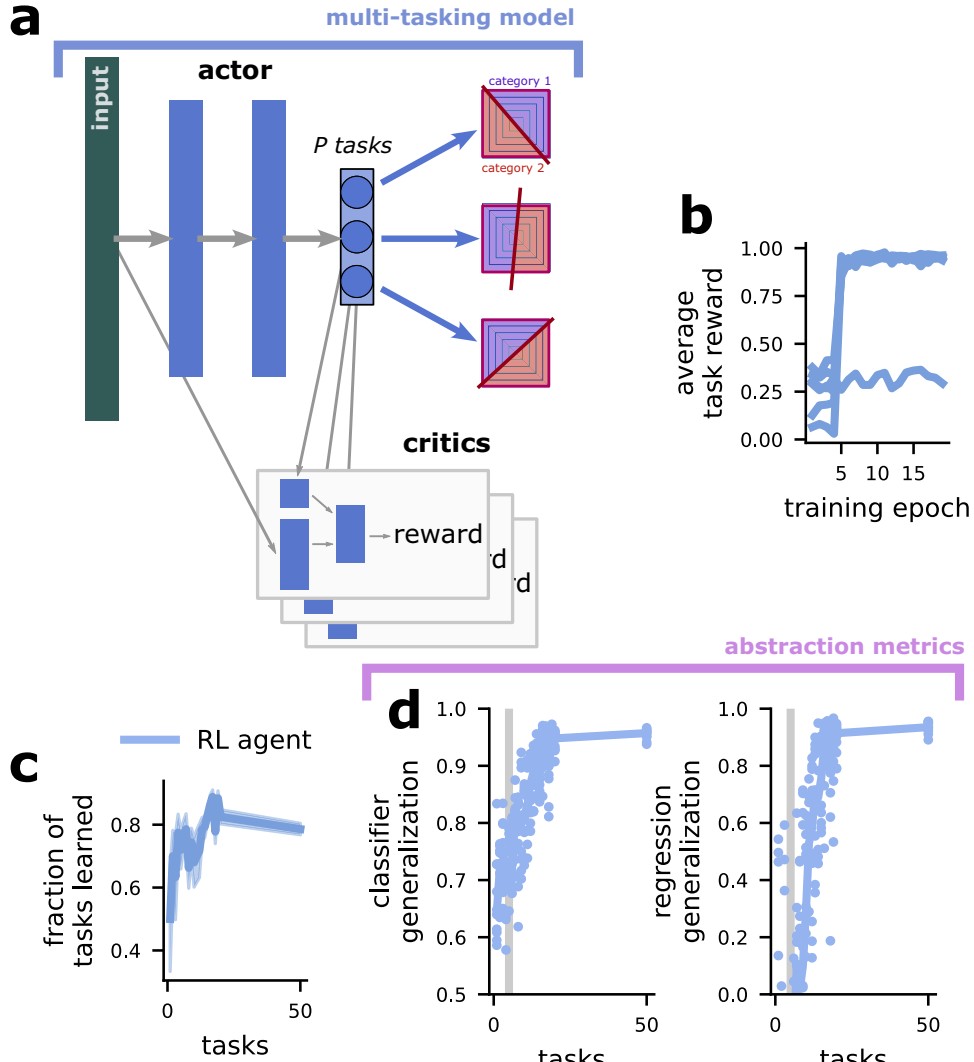

**Fig. 7 | The multi-tasking model produces abstract representations when trained with reinforcement learning. a** Schematic of the reinforcement learning multi-tasking model, using the deep deterministic policy gradient approach. **b** The performance of the network on different tasks over the course of training. Note the sharp transitions between near-chance performance (avg reward = 0) and near-perfect performance (avg reward = 1). **c** The fraction of tasks learned such that avgreward > 0.8 for different numbers of trained tasks, the shading is the standard error of the mean and there are $n = 10$ models included in the analysis. **d** The classifier (left) and regression (right) generalization performance of the reinforcement learning multi-tasking model.

reliable. They are learned even for heterogeneous classification tasks, stimuli with partial information, in spite of being required to learn additional non-latent variable-aligned tasks, and for a variety of tangled, high-dimensional, and image inputs. Finally, we show that the multi-tasking model develops abstract representations even when trained with reinforcement rather than supervised learning. Overall, this work provides insight into how the abstract neural representations characterized in experimental data may emerge: Through the multiple constraints and complexity induced by naturalistic behavior.

Representations in the brain are often observed to be sparse[26]. Here, while the standard input and RF input (fig. S4) that we explore are highly sparse, the abstract representations that the multi-tasking model develops are not necessarily sparse. Indeed, when we characterize the sparseness of representations in the multi-tasking model, we find that they are substantially less sparse than the inputs (fig. S9a, c, left). To explore this apparent inconsistency, we apply regularization to the activity in the representation layer of the multi-tasking model. In models trained with weak $L_1$ and $L_2$ regularization, we find only a small decrease in the classification and regression generalization performance (fig. S9b, d) along with a striking increase in the average

sparseness across the population (though it remains less sparse than the input, fig. S9a, c). Thus, sparseness and abstract representations can coexist in the multi-tasking model. Further, the representation of facial features in the brain is thought to share this property: In the whole population of inferotemporal cortex neurons, face selectivity is relatively rare – and so the representation is sparse[48] (though face cells are also concentrated in particular anatomical subdivisions of the inferotemporal cortex[49]). However, within face-responsive neurons, the code is almost linear in facial features[9] and is abstract[11,20]. We can view this as two hierarchical codes. The outer code is a sparse representation of object identity (e.g., face or hand). The inner code is a dense, abstract code for the features of that object (e.g., a happy or sad expression). This may be a general strategy for object representations in the primate brain[50]. Further, this particular kind of sparse representation has been explored in machine learning[51–53] and is thought to be essential for flexible and intelligent behavior[54].

While we find fully abstract representations for the standard input (Fig. 3), receptive field inputs (fig. S4), and image inputs (Fig. 6), we do not find fully abstract representations for low length scale random Gaussian process inputs (Fig. 5f, g). The low length scale random

Gaussian process input differs from all other input types in one important way: Both linear decoders and regressions perform relatively poorly even when trained and tested on the whole stimulus space (Fig. 5d). Thus, this initial linear separability may be a prerequisite for the multi-tasking model to produce abstract representations. Further, it suggests that a crucial step may be an initial dimensionality expansion, that produces this separability, before the dimensionality of the representation is collapsed again into an abstract form. Future work will investigate incorporating this into the multi-tasking model through regularization of the first layer.

We train our models to perform different binary classifications of latent variables as a proxy for different behaviors. This is, of course, a highly simplified approach. While feedforward binary classification most closely matches rapid object recognition or, for example, go or no-go decisions, it does not provide an accurate model of behaviors that unfold over longer timescales. While most of the experimental work that shows abstract representations in the brain[8–10,12] and other models that produce abstract representations in machine learning systems[18,21,22] have taken a static view of neural activity, network dynamics could play a role in establishing and sustaining abstract representations. Interestingly, recent work has shown that neural networks trained to predict the result of a chosen action develop low-dimensional, potentially abstract representations of the latent space underlying the observations[55]. This form of prediction could be viewed as a multi-tasking problem similar to the one we studied here—and could indicate that abstract representations may emerge naturally from predicting the sensory consequences of our actions, without explicit feedback.

In addition to a potential role for dynamic prediction in producing abstract representations, there is growing literature on the ability of network dynamics to implement abstract operations. In particular, recent work has shown that training recurrent neural networks to perform multiple dynamic tasks leads to shared implementations of common task operations (such as storing information across a delay period)[30–32]. As a result, novel tasks can be quickly acquired through the combination of these learned abstract operations[31]. This is an important form of abstraction that differs from the abstract representations we have studied here. We believe that the two forms can work in tandem: Abstract representations (in our sense) may be important for the abstract operations to be robust to irrelevant changes in context. However, our work suggests that these abstract representations may emerge naturally from the multi-task training that these networks already undergo. We believe that further work can fruitfully combine these two lines of research.

Our method of quantifying abstractness in both artificial and biological neural networks has an important difference from some previously used methods[10]. In particular, an influential model for creating disentangled representations in machine learning, the $\beta$ variational autoencoder ($\beta$VAE, and see "Comparing the multi-tasking model with the unsupervised $\beta$VAE" in Supplementary Methods), attempts to isolate the representation of single latent variables to single units in the network[18]. Directly applied to neural data, this leads to the prediction that single neurons should represent single latent variables in abstract representations[10]. These single-neuron representations of single latent variables lead to distinct modules within the neural population, one module for each latent variable. This kind of representation would also be abstract under our metrics, and can be viewed as a special case in which the axes of neural population space are aligned with the latent variables. Our abstraction metrics, however, do not require this alignment. They depend on the geometry of the representations at the population level and this geometry is unaffected by whether single neuron activity corresponds to a single latent variable, or to a linear mixture (i.e., a weighted sum) of all the latent variables. Given the extensive linear and nonlinear mixing observed already in the brain[4,8,9,56], we believe that this flexibility is an advantage

of our framework for detecting and quantifying the abstractness of neural representations. Further, we believe that searching for abstract representations using techniques that are invariant to linear mixing will reveal abstract representations where they may not have been detected previously—in particular, a representation can provide perfect generalization performance without having any neurons that encode only a single latent variable, and thus such a representation would not be characterized as abstract by many machine learning abstraction or disentanglement metrics.

For experimental data, our findings predict that an animal trained to perform multiple distinct tasks on the same set of inputs will develop abstract representations of the latent variable dimensions that are used in the tasks. In particular, if the tasks only rely on three dimensions from a five-dimensional input, then we expect strong abstract representations of those three dimensions (as in fig. S2c, d), but not of the other two. We expect all of the dimensions to still be represented in neural activity, however,—we just do not expect them to be represented abstractly. Once this abstract representation is established through training on multiple tasks, if a new task is introduced that is aligned with these learned latent variables, we expect the animal to be able to learn and generalize that task more quickly than a task that relies on either the other latent variables or is totally unaligned with the latent variables (as the grid tasks above). That is, we expect animals to be able to take advantage of the generalization properties provided by abstract representations that we have focused on throughout this work, as suggested by previous experimental work in humans[36].

A recent study in which human participants learned to perform two tasks while in a functional magnetic resonance (fMRI) scanner provides some evidence for our predictions[15]. The representations of a high-dimensional stimulus with two task-relevant dimensions (one which was relevant in each of two contexts) were studied in both the fMRI imaging data and in neural networks that were trained to perform the two tasks (the setup in this work is similar to certain manipulations in our study, particularly to the partial information case shown in fig. S2a, b). They find that the representations developed by a neural network that develops rich representations (similar to abstract representations in our parlance) are more similar to the representations in the fMRI data than neural networks that develop high-dimensional, non-abstract representations. This provides evidence for our central prediction: That abstract representations emerge through multiple-task learning. However, the conditions explored in the human and neural network experiments in the study were more limited than those explored here. In particular, only two tasks were performed, the stimulus encoding was less nonlinear than in our studies, and the tasks were always chosen to be orthogonal. Thus, further work will be necessary to determine the limits of our finding in real brains.

Several additional predictions can be made from our results with the grid tasks, which showed that learning many random, relatively uncorrelated tasks both does not lead to the development of abstract representations alone, but also does not interfere with abstract representations that are learned from a subset of tasks that are aligned with the latent variables. First, if an animal is trained to perform a task analogous to the grid task, then we do not expect it to show abstract representations of the underlying latent variables – this would indicate that latent variables are not inferred when they do not support a specific behavior. Second, we predict that an animal trained to perform some tasks that are aligned to the latent variables as well as several (potentially more) non-aligned grid task analogs will still develop abstract representations. Both of these predictions can be tested directly through neurophysiological experiments as well as indirectly through behavioral experiments in humans (due to the putative behavioral consequences of abstract representations[36]).

Overall, our work indicates that abstract representations in the brain – which are thought to be important for generalizing knowledge

across contexts – emerge naturally from learning to perform multiple categorizations of the same stimuli. This insight helps to explain previous observations of abstract representations in tasks designed with multiple contexts (such as ref. [8]), as well as makes predictions of conditions in which abstract representations should appear more generally.

## Methods
### Abstraction metrics
Both of our abstraction methods quantify how well a representation that is learned in one part of the latent variable space (e.g., a particular context) generalizes to another part of the latent variable space (e.g., a different context). To make this concrete, in both metrics, we train a decoding model on representations from only one−randomly chosen −half of the latent variable space and test that decoding model on representations from the non-overlapping half of the latent variable space.

**The classifier generalization metric**. First, we select a random balanced division of the latent variable space. One of these halves is used for training, the other is used for testing. Then, we select a second random balanced division of the latent variable space that is orthogonal to the first division. One of these halves is labeled category 1 and the other is labeled category 2. As described above, we train a linear classifier on this categorization using 1000 training stimuli from the training half of the space, and test the classifier's performance on 2000 stimuli from the testing half of the space. Thus, chance is set to 0.05 and perfect generalization performance is 1.

**The regression generalization metric**. As above, except we train a linear ridge regression model to read out all $D$ latent variables using 4000 sample stimulus representations from the training half of the space. We then test the regression model on 1000 stimulus representations sampled from the testing half of the space. We quantify the performance of the linear regression with its $r^2$ value:

$$r^2 = 1 - \frac{\text{MSE}(X, \hat{X})}{\text{Var}(X)} \tag{4}$$

where $X$ is the true value of the latent variables and $\hat{X}$ is the prediction from the linear regression. Because the MSE is unbounded, the $r^2$ value can be arbitrarily negative. However, chance performance is $r^2 = 0$, which would be the performance of the linear regression always predicted the mean of $X$, and $r^2 = 1$ indicates a perfect match between the true and predicted value.

### Non-abstract input generation
In the main text, we use two methods for generating non-abstract inputs from a $D$-dimensional latent variable. We have also performed our analysis using several other methods, which we also describe here.

**Participation ratio-maximized representations**. We train a symmetric autoencoder (layers: 100, 200 units) to maximize the participation ratio[38] in its 500 unit representation layer. The participation ratio is a measure of embedding dimensionality that is roughly equivalent to the number of principal components that it would take to capture 80% of the total variance. The autoencoder ensures that information cannot be completely lost, while the participation ratio regularization ensures that the representation will have a high-embedding dimension and, therefore, be non-abstract. The performance of our generalization metrics on this input representation is shown in Fig. 1f.

**Random Gaussian process inputs**. To generate the random Gaussian process inputs, we proceed through each input dimension separately. For each dimension, we sample a single $D$-dimensional function from

the prior of a Gaussian process with a radial basis function kernel of length scale $l$. Then, the full input is simply the vector of all of these input dimensions.

We use the implementation of Gaussian processes provided in scikit-learn[57]. In particular, we initialize a Gaussian process with the above kernel, then take a sample scalar output from the Gaussian process prior distribution for a selection of 500 random points. Then, we freeze this function in place by fitting the Gaussian process to reproduce this output sample from the same set of input points.

### Quantifying sparseness and dimensionality
Throughout the paper, we use a standard per-unit measure of sparseness[26,58–60],

$$S = 1 - \frac{E[r(\mathbf{x})]_X^2}{E[r(\mathbf{x})^2]_X} \tag{5}$$

where $r(\mathbf{x})$ is the response of a particular unit to input $\mathbf{x}$. We have neglected the usual normalization by $1-1/n$ where $n$ is the number of stimuli because we sample thousands of stimuli. The measure ranges from 0 to 1 and is close to 1 when the unit primarily responds to one stimulus or a few stimuli; if all the stimuli have similar firing, then the measure is close to zero.

We also use the participation ratio[38] to quantify the embedding dimensionality of neural representations, which can also be viewed as a measure of sparseness across the population for rectified linear units. The participation ratio is defined as follows,

$$PR = \frac{(\sum_i^N \lambda_i)^2}{\sum_i^N \lambda_i^2} \tag{6}$$

where $\lambda_i$ are the eigenvalues of the population response across $N$ units. The participation ratio is 1 if there is only one non-zero eigenvalue and $N$ if all $N$ eigenvalues are the same magnitude. In intermediate regimes, it can be viewed roughly as the number of dimensions necessary to explain 80% of the population variance[38].

### The multi-tasking model
We primarily study the ability of the multi-tasking model to produce abstract representations according to our classification and regression generalization metrics. The multi-tasking model is a feedforward neural network. For Figs. 3 and 4 it has the following parameters:

| | |
|---|---|
| layer widths | 250, 150, 100, 50 |
| representation width | 50 |
| batch size | 100 |
| training examples | 10000 |
| epochs | 200 |

For fig. S4, everything is kept the same except the number of layers is increased:

layer widths| 500, 250, 50

The increased number of layers improves performance in the RF case. However, for the standard input (and the images, as described below), the results are similar with only three layers (see A sensitivity analysis of the multi-tasking model and $\beta$VAE in Supplementary Methods).

**Full task partitions**. In all cases, the models are trained to perform multiple tasks−specifically, binary classification tasks−on the latent variables. In the simplest case (i.e., Fig. 3e), the task vector can be

written as,

$$T(\mathbf{x}) = \text{sign}\, A\mathbf{x} \tag{7}$$

where $A$ is a $P \times D$ matrix with randomly chosen elements and $\mathbf{x}$ is the $D$-dimensional stimulus.

**Unbalanced task partitions.** For unbalanced partitions, the task vector has the following simple modification,

$$T(\mathbf{x}) = \text{sign}\,[A\mathbf{x} + b] \tag{8}$$

where $b$ is a $P$-length vector and $b_i \sim \mathcal{N}(0, \sigma_{\text{offset}})$. Notice that this decreases the average mutual information provided by each element of $T(\mathbf{x})$ about $\mathbf{x}$.

**Contextual task partitions.** We chose this manipulation to match the contextual nature of natural behavior. As motivation, we only get information about how something tastes for the subset of stimuli that we can eat. Here, we formalize this kind of distinction by choosing $P$ classification tasks that each only provide information during training in half of the latent variable space.

We can write each element $i$ of the contextual task vector as follows,

$$T_i(\mathbf{x}) = \begin{cases} \text{sign}\,[A_i\mathbf{x} + b_i] & C_i\mathbf{x} > 0 \\ \text{nan} & C_i\mathbf{x} \leq 0 \end{cases} \tag{9}$$

where nan values are ignored during training and $C$ is a $P \times D$ random matrix. Thus, each of the classification tasks influences training only within half of the latent variable space. This further reduces the average information provided about $\mathbf{x}$ by each individual partition.

**Partial information task partitions.** For contextual task partitions, the contextual information acts on particular tasks. For our partial information manipulation, we take a similar structure, but it instead acts on specific training examples. The intuitive motivation for this manipulation is to mirror another form of contextual behavior: At a given moment (i.e., sampled training example) an animal is only performing a subset of all possible tasks $P$. Thus, for a training example from that moment, only a subset of tasks should provide information for training.

Mathematically, we can write this partial information structure as follows. For each training example $\mathbf{x}$, the task vector is given by,

$$T_i(\mathbf{x}) = \begin{cases} \text{sign}\,[A_i\mathbf{x} + b_i] & p \geq M \\ \text{nan} & p < M \end{cases} \tag{10}$$

where $p$ is a uniformly distributed random variable on $[0, 1]$, which is sampled uniquely for each training example $\mathbf{x}$ and $M$ is a parameter also on $[0, 1]$ that sets the fraction of missing information. That is, $M = 0.9$ means that, for each training example, 90% of tasks will not provide information.

While results are qualitatively similar for many values of $M$, in the main text we use a stricter version of this formalization: For each training sample, one task is randomly selected to provide information, and the targets for all other tasks are set to nan.

**Grid classification tasks.** The grid tasks explicitly break the latent variable structure. Each dimension is broken into $n$ parts with roughly equal probability of occurring (see schematic in Fig. 4a). Thus, there are $n^D$ unique grid compartments, each of which is a $D$-dimensional volume in latent variable space, and each compartment has a roughly equal probability of being sampled. Then, to define classification tasks on this space, we randomly assign each compartment to one of the two categories – there is no enforced spatial dependence.

**Random Gaussian process tasks.** To generate a random Gaussian process task indexed by $i$, we sample a single $D$-dimensional function from the prior of a Gaussian process with a radial basis function kernel of length scale $l$, which we denote as $\text{GP}_i^l$. Then, to determine the category of a particular sample $\mathbf{x}$, we evaluate the function on that category,

$$T_i(\mathbf{x}) = \text{sign}\,(\text{GP}_l(\mathbf{x})) \tag{11}$$

We use the implementation of Gaussian processes provided in scikit-learn[57]. In particular, we initialize a Gaussian process with the above kernel, then take a sample scalar output from the Gaussian process prior distribution for a selection of 500 random points. Then, we freeze this function in place by fitting the Gaussian process to reproduce this output sample from the same set of input points.

**The dimensionality of representations in the multi-tasking model.** First, we consider a deep network trained to perform $P$ balanced classification tasks on a set of $D$ latent variables $X \sim \mathcal{N}(0, I_D)$. We focus on the activity in the layer just prior to readout, which we refer to as the representation layer and denote as $r(\mathbf{x})$ for a particular $\mathbf{x} \in X$. This representation layer is connected to the $P$ output units by a linear transform $W$. In our full multi-tasking model, we then apply a sigmoid nonlinearity to the output layer. To simplify our calculation, we leave that out here. The network is trained to minimize error, according to a loss function which can be written for a particular sample $\mathbf{x}$ as:

$$L(x) = \frac{1}{2}\sum_i^P [\text{sign}\,(A\mathbf{x}) - Wr(\mathbf{x})]_i^2 \tag{12}$$

where $A$ is a $P \times D$ matrix of randomly selected partitions (and it is assumed to be full rank) and the sum is over the $P$ tasks. To gain an intuition for how $r(\mathbf{x})$ will change during training, we write the update rule for $r(\mathbf{x})$ (to be achieved indirectly by changing preceding weights, though we ignore the side effects that would arise from these weight changes),

$$r(\mathbf{x})^{s+1} = r(\mathbf{x})^s - \mu \frac{\partial L}{\partial r(\mathbf{x})} \tag{13}$$

$$= r(\mathbf{x})^s + \mu W^T \text{sign}\,(A\mathbf{x}) - \mu W^T W r(\mathbf{x}) \tag{14}$$

Thus, we can see that, over training, $r(\mathbf{x})$ will be made to look more like a linear transform of the target function, $\text{sign}(A\mathbf{x})$. Next, to link this to abstract representations, we first observe that $A\mathbf{x}$ produces an abstract representation of the latent variables. Then, we show that $\text{sign}(A\mathbf{x})$ has approximately the same dimensionality as $A\mathbf{x}$. In particular, the covariance matrix $M = E_X\left[\text{sign}(A\mathbf{x}\mathbf{x}^T A^T)\right]$ has the elements,

$$M_{ij} = 1 - \frac{2}{\pi}\arccos A_i A_j \tag{15}$$

where $A_i$ is the $i$th row of $A$ and when $x_i$ are random variables with an equal probability of being positive or negative (both the Gaussian and uniform distributions we use here have this property). To find the dimensionality of $\text{sign}(A\mathbf{x})$ we need to find the dimensionality of $M$. First, the distribution of dot products between random vectors is centered on 0 and the variance scales as $1/D$, where $D$ is the dimensionality of the latent variables as usual. Thus, we can Taylor expand

the elements of the covariance matrix around $A_i A_j = 0$, which yields

$$M_{ij} \approx \frac{2}{\pi} A_i A_j \qquad (16)$$

We identify this as a scalar multiplication of the covariance matrix for the linear, abstract target $E_X[A\mathbf{x}\mathbf{x}^T A^T]$. Further, we know that the rank of this matrix is min($P,D$). So, this implies that the matrix $M$ also has rank approximately min($P,D$). Deviations from this approximation will produce additional non-zero eigenvalues, however, they are expected to be small.

Importantly, while a high dimensional $r(\mathbf{x})$ can solve $P$ classification tasks in a non-abstract way (for example, notice that the classification accuracy of the standard inputand RF inputs below have very high classification accuracy for random tasks yet much lower generalization performance, Fig. 1f, left and fig. S4b, left), an $r(\mathbf{x})$ with dimensionality min($P,D$) will be constrained to solve the tasks in at least partially abstract way (see "Four possibilities for representations in the multi-tasking model" in Supplementary Methods).

### Pre-processing using a pre-trained network
When applying the multi-tasking model to image inputs, we used a deep neural network trained on the ImageNet classification task to pre-process them into a feature vector. Then, we used this representation as input to the multi-tasking model. The pre-trained network is not fine tuned, or trained, during the training of the multi-tasking model.

The parameters of the model used here are the same as the multi-tasking model except:

layer widths| 250, 50

and only 300 of the unique chair types were used in the training dataset.

The network we used is available here: https://tfhub.dev/google/efficientnet/b0/feature-vector/1.

### The image datasets
We used two standard image datasets from the machine learning literature. In both cases, we considered a subset of the total number of features, explained below.

**The 2D shapes dataset**. This dataset consists of white 2D shapes on a black background[43]. The features are horizontal and vertical position, 2D rotation, scale, and shape type. We do not train tasks on either the rotation or shape type variables. We exclude rotation due to its periodicity and shape type because it is a categorical variable. All values of both variables are still included in the training dataset, they are simply not used in the classification tasks.

**The 3D chair dataset**. This dataset consists of images of different styles of chairs on a white background[42]. The original features are azimuthal rotation, pitch, and chair style. We augment these features to include horizontal and vertical position by translating the image using coordinates sampled from a normal distribution and truncated when the chair portion of the image begins to wrap around the edges. We exclude pitch, a subset of azimuthal rotations, and chair styles from task training. We exclude pitch because it has only two values in the dataset and chair style because it is a categorical variable. Both are still included in the training data. We exclude a subset of azimuthal rotations to break the periodicity of the variable, which allows us to treat azimuthal rotation as a continuous, non-periodic variable.

### The reinforcement learning multi-tasking model
We adapt the DDPG[47] to train the multi-tasking model. In particular, we train a single-actor network, which is tasked with taking in a stimulus and producing an action. The action is the categorization of that stimulus on each of the $P$ trained tasks. To provide supervision to this actor network, we train independent critic networks for each task, which take in the stimulus and the action produced by the actor and attempt to predict the reward that will be received from that pair. These critic networks are trained with respect to the actual reward received, and the predicted reward from the critic is used to train the actor network. Otherwise, we follow the standard DDPG approach as described in ref. [47].

The correct action for a categorization task was 1 for one category and −1 for the other category. An action was rewarded if the network produced positive activity greater than the reward/punishment threshold for the former and negative activity greater than that threshold for the latter. If the activity was greater than that threshold but with the wrong sign, then the network received a punishment (i.e., a negative reward). Otherwise, no reward or punishment was received.

The parameters of the reinforcement learning multi-tasking model are:

| | |
|---|---|
| Actor layer widths | 250, 150, 50 |
| Representation width | 50 |
| Critic stimulus layer widths | 10, 5 |
| Critic action layer widths | 1 |
| Critic shared layer widths | 5 |
| Batch size | 200 |
| Training epochs | 50119 |
| Initial batches | 20000 |
| Reward/punishment threshold | ±0.33 |

### The βVAE
The βVAE is an autoencoder designed to produce abstract (or, as referred to in the machine learning literature, disentangled) representations of the latent variables underlying a particular dataset[18]. The βVAE is totally unsupervised, while the multi-tasking model receives the supervisory task signals. Abstract representations are encouraged through tuning of the hyperparameter $\beta$, which controls the strength of regularization in the representation layer, which penalizes the distribution of representation layer activity for being different from the standard normal distribution. In fig. S3, the βVAE is trained with the same parameters as given in section M6− the layers are replicated in reverse for the backwards pass through the autoencoder. For fig. S4, the parameters are as described in section M6. In both cases, instead of fitting models across different numbers of partitions, we fit the models with different values chosen for $\beta$.

For fig. S5, parameters for the βVAE are as described in section S4.1. We also explored numerous other architectures for the βVAE in that figure, but never obtained qualitatively or quantitatively better results.

### Reporting summary
Further information on research design is available in the Nature Portfolio Reporting Summary linked to this article.

## Data availability
The large-scale simulation data generated in this study have been deposited in the Figshare database and at the following link: https://doi.org/10.6084/m9.figshare.21761348.v1. More detail about how to use these data to generate the figures is provided in this github repository: https://github.com/wj2/disentangled.

## Code availability

All of our code for this project is written in Python, making extensive use of TensorFlow[61] and the broader python scientific computing environment (including numpy[62], scipy, matplotlib, and scikit-learn[57]). The code is available in the follow repository: https://github.com/wj2/disentangled. The version of the code used to generate these figures is here: https://doi.org/10.5281/zenodo.7465963.

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

## Acknowledgements

We thank Mattia Rigotti, Nicolas Masse, and Matthew Rosen for their comments on an earlier version of this manuscript. This work was supported by the following grants: Simons Foundation 542983SPI (S.F. and W.J.J.), Neuronex NSF 1707398 (S.F. and W.J.J.), Gatsby Charitable Foundation GAT3708 (S.F. and W.J.J.), and the Swartz Foundation (S.F. and W.J.J.). We also thank Allison Ong, Aleyna Silcott, and Mahham Fayyaz for administrative support. We acknowledge computing resources from Columbia University's Shared Research Computing Facility project, which is supported by NIH Research Facility Improvement Grant 1G20RR030893-01, and associated funds from the New York State Empire State Development, Division of Science Technology and Innovation (NYSTAR) Contract C090171, both awarded 15 April 2010.

## Author contributions

W.J.J. and S.F. conceived the project and developed the simulations. W.J.J. performed the simulations and analytical calculations. W.J.J. analyzed the simulation results and made the figures. W.J.J. and S.F. wrote and edited the paper.

## Competing interests

The authors declare no competing interests.
