## [Peer Review File · Nature Communications]

Abstract representations emerge naturally in neural networks trained to perform multiple tasksReviewers' Comments:

Reviewer #1:

Remarks to the Author:

This paper analyzes and compares the geometry of high-level/latent representations of artificial neural networks trained on multiple binary stimulus classification tasks. The key result is that the latent representations become more abstract/linear as the number of classification tasks increases. While the paper is not directly examining the geometry of neural representations obtained from neural recordings, the importance of the findings relate to the ability of representations to generalize or not to novel stimuli and tasks. They first look at artificially created latent representations and the high-level/latent representations of feedforward ANNs (generally the representations of the last layer of a feed-forward network before the output layer). Geometry is quantified by the ability of the classifier to generalize to novel instances of the stimulus set, and by quantifying whether a linear regression on the hidden representations generalizes to a novel stimuli. Using a number of different types of networks they observe that the representations become more abstract/linear as the number of tasks increases. Thus one of the predictions made, is that the abstractness of neural representations in animals should increase with the number of tasks the animal is trained on.

The geometry of representations is an important, and increasingly studied, problem in neuroscience and machine learning. In ML it is well established that effective word embeddings are abstract and as a result of their linear properties allow for "word algebra". The standard example being that the representation of KING - MAN + WOMAN \approx QUEEN; and that these embeddings benefit from large datasets (not that words correspond to directly to number of tasks). In the ML field the findings are probably expected, although some of the methodologies may be novel. But it may be helpful to determine if such representational algebra applies to the stimulus classes studied here.

I found the paper rather difficult to follow and I suspect other readers will as well. Part of the confusion arises because of the multiple meanings of "model" which can refer to the feedforward network or the decoders used to analyze the representations in the feedforward network. Whether the network or the decoders are being trained on multiple tasks also is a source of confusion. Additionally, some panels are called out of order (e.g., 2a is called before 1f), and I don't think Figs. S1 S5 S6 S7 S8 are mentioned in the main text. In the early description of the Results, the explanation of the methods is not very clear (e.g. clarify which latent representations the two abstraction metrics are performed on). Finally, too much critical information needed understand the early figures is referred to the Methods, making it impossible to understand the Results without first reading the Methods very carefully. For example, in the first paragraph of Section 2.2 there are three calls to the Methods and Supplements, making it hard to understand the key results of Fig 2 without a lot of back and forth.

Here the latent representations were high-dimensional and non-sparse. Neural representations sometimes appear to be quite sparse (e.g., in the ventral visual stream). Do the current results hold for more sparse representations?

Are the results dependent on the "dimensionality" (number of units in the representation) of the representations? It would seem that the higher the dimensionality the larger the number of tasks would have to be for the linearity to arise?

It is not clear if the clear linear/abstract representations are in part dependent on the assumption used, including the linear decoding and the regularization. That is, is it necessarily the case that if you are using a linear decoder you inevitably get linear representations as the number of tasks increases? Is there a mathematical theorem that states that since one is reducing the error during training using standard loss functions, demonstrates that the linearity of the representations must increase as one trains more and more tasks using linear readouts?

In Fig 6c why is it that the fraction of tasks learned start low go up and come down? Is this related to abstract representations? Provide the n's and SEM for the curve in the panel.

The term "dimensionality" is a bit confusing. Generally, there are several types of dimensionality: the intrinsic dimensionality—the intrinsic geometric properties of a given manifold or latent space, such as a 2D sphere; the extrinsic dimensionality of a manifold dependent on how the manifold is embedded in the ambient space; and the dimensionality of the ambient space itself. The author might want to clarify such a definition whenever it is referred.

Reasonably so, the authors focus on the last layer for the representation since it is closest to the output. The reader might also want to know how such representations evolve through different layers, in other words, is there a progressive development of abstract representations through layers and is it also dependent on the number of tasks.

Sections 2.5 and 2.6 discuss the same figure, but called the panels in a confused way: the results were referred in 2.5 before it was explained how it was generated up to section 2.6.

Fig 3, panels b and d are confusing, it looks like the right panels are part of the same panel. Having separate y axis labels might help.

There are a number of typos, which are a bit hard to list in the absence of line numbers in the manuscript. For example:

Pg. 7 "an multi-taking model"

Pg. 15 "rapid objection recognition"

Pg. M2 "we we"

Reviewer #2:

Remarks to the Author:

The authors show that when neural networks are trained to perform multiple tasks that rely on linearly classifying latent variables, the representations that emerge are linear embeddings of the latent variables. These linear embeddings are an efficient substrate for learning and, in particular, allow a classifier or regression model trained on one region of latent space to generalize to other regions of latent space.

I thought this was an interesting, well-executed study that seeks to address an important question. My major concerns were about the way in which the task structure naturally imposed the desired representation and thus whether the movement from task structure to representation really captures what we mean by abstraction. I have elaborated this and other concerns below:

1) The study relies quite heavily on linearity—the notion of abstract representations is that they are linearly represented in neural population activity space and the tasks that the network is trained on correspond to linear separation in latent variable state. Given that the final transformation from representation layer to readout layer is near linear, as far as I understand, the theoretical argument is that if one trains a network to read out (at least) D linear functions of D variables using linear projections, then the representation must necessarily be linear in the D variables.

I think the first use (abstract representation=linear) is an interesting formulation of abstractness but could be unpacked a bit more and contrasted to other notions of "abstract" (besides the beta-VAE). For example, how does it relate to notion of abstraction as extracting modular, compositional structure from the world? I'm guessing that the implicit assumption is that the linearly separable factors are the

modules that can be flexibly composed but would be nice to have this explicit.

2) I have concerns with the second use of linearity from the point above—the close match between the desired representations and the trained tasks is a strong assumption. In essence, it seems like each task causes the network to learn a linear representation along one dimension in latent space and D such independent tasks force abstraction (or linearity) along D dimensions (though I appreciate the authors' point that this is not the only solution that can be conceived of). This way of abstracting seems quite different from a more usual notion where each single or small subset of tasks can be performed without abstraction but an organism or algorithm learns to construct abstract representations because they are useful and efficient across distinct tasks. It also seems distinct from learning modular computations, representations or dynamics that can be reused across tasks (e.g., the Yang 2019 paper). If this is a fair characterization, I think the paper should at minimum include some more discussion of the role this match between desired representations and task structure plays.

Note that the existing simulations of contextual and partial information task partitions and the more complex GP task boundaries go some way to addressing my concerns, and make me wonder if there's some picture in which multiple locally linear task boundaries are averaged or stitched together across multiple tasks to get global linearity. If true I think this would be very interesting and could be highlighted as a way in which multiple tasks are fundamentally important.

3) Related to the above, the novel tasks are set up so that they can be learned as a one layer linear readout from the D dimensional representation so I would expect the performance to be identical to learning directly from the latent variables (as the authors show). Do the results hold for a more complex novel task, which for example might require tuning weights in earlier layers? Maybe the GP task boundaries from later in the study could be used?

4) Also related to the linearity constraint, the authors often say high dimensionality when they mean nonlinearity / high embedding dimension. E.g., "high-dimensional representation often do not permit generalization" might be true, but the claim here is about low-dimensional but nonlinear representations. The authors are of course aware of the distinction but it should be made clear which is meant throughout.

5) In the sensitivity analysis, does the shrinking size of deeper layers play an important role? Would the network learn the low-dimensional representation (as opposed to just copying the standard input) if deeper layers were the same size? It looks from Fig. S7 like doubling the size of just the representation layer doesn't have an effect but is a bottleneck at some point important?

6) Sections 2.5 and 2.6 were confusing and took me several reads (though the analyses themselves are really nice and I appreciate the authors' thoroughness). I think this is entirely because of how the figure is organized and the text itself is relatively clear (though 2.6 could be clarified a bit more). Specifically, as far as I can tell, Section 2.5 is about using curved boundaries in output task space rather than linear ones, and Section 2.6 is about using a different parameterization of the input (i.e., GP tuning curves) but the corresponding figure is interleaved between sections and in particular Fig. 4a goes with section 2.6. I would suggest just splitting into two smaller figures that go with the appropriate section, or if the authors must keep a big figure, to put anything related to Section 2.5 before 2.6.

7) The lower performance for the GP input and the receptive field input seems like an important issue, given that GP tuning curves and localized receptive fields are closer to biology than the autoencoder. Up until this section, my read was that the autoencoder used to generate the standard input just generated some highly nonlinear transformation of the input that was then unlearned by the multitasking model. But these results suggest that there's something special about the autoencoder input. Can performance like the autoencoder input be recovered with these more natural input structures and for what depth of network or objective function does this happen? If not there might be

an interesting scientific point to be made here about the need for a two-stage sensory/cognitive model (with a separate loss function for the sensory representation), especially given the results in Section 2.7 that training the multitasking model on top of an object recognition model yields abstraction.

Minor comments:

8) The paper gets to geometry, nonlinearity of representation, high embedding dimension, etc. quite quickly (e.g., "Neural representations of sensory and cognitive variables are often highly nonlinear and have high embedding dimension"). For accessibility to non-theorist audiences this should be unpacked a bit, even if only a paragraph quickly summarizing the geometric view of neural coding.

10) Pg. 4 "Feedback onto representation layer" I'm guessing feedback through learning since it's a feedforward network? If so "feedback" is confusing.

11) What are gray lines in Fig. 2b and c?

12) Equation at top of Pg. 6: I guess x is input, $r(x)$ is value in representation layer and W is connections in last layer. But this should all be defined and argument unpacked slightly so argument can be read without looking ahead to Methods.

13) The linear representation is necessarily dense, with most neurons active for most inputs. How do the authors see this as comparing to sparse activation in neural populations?

14) Other recent work has found that low-dimensional linear (i.e., abstract) structure emerges in network that are trained to perform predictive learning (Recanatesi et al. 2021). Is there a relationship between these frameworks? If so making that explicit would be useful to the field.

15) Note repeated sentence on Pg. S-1 and S-4 "However, we believe the contrast is still informative..."

Response to the reviewers

We appreciate the detailed and thoughtful comments from both reviewers. By addressing these comments, we believe that we have significantly strengthened the paper, both scientifically and stylistically. We have changed the manuscript in three main ways:

1. We have included a new figure that clearly delineates the different parts of our framework: from the input model, to the multi-tasking model, to the abstraction metrics we use throughout. We also use this figure to introduce a new color scheme for each of these three central concepts in the paper, and have propagated that color scheme through the rest of the main text figures. In addition to these visual changes, we made the introduction of each concept and their interrelation more complete in the main text. These changes are all described in detail below. We have also worked to improve clarity by including some new visualizations of our findings.
2. We have included extensive new computational experiments to address different questions and concerns raised by the reviewers. We have included an example showing that the abstract representations developed here can be used in a compositional manner, to generate images with specific properties (fig. S5). Other additional experiments include: simulations with regularization to encourage sparse representations in the multi-tasking model (fig. S9), simulations that show the insensitivity of our findings to changes in the embedding dimensionality of the input (fig. S10), and simulations that show the insensitivity of our findings to the width of the layers in the multi-tasking model (fig. S11).
3. We have also extensively rewrote the manuscript to improve the clarity about our methods as well as our results.

Through these changes and new inclusions, we believe that we have been able to substantively address the reviewers' concerns.

Below, we address each of the specific concerns of the reviewers in the order they were provided in the reviews. We reproduce the text written by the reviewers in blue and include much of our new text here in green. All line numbers refer to the clean version of the new manuscript.

1 Review 1

1.1 Summary

This paper analyzes and compares the geometry of high-level/latent representations of artificial neural networks trained on multiple binary stimulus classification tasks. The key result is that the latent representations become more abstract/linear as the number of classification tasks increases. While the paper is not directly examining the geometry of neural representations obtained from neural recordings, the importance of the findings relate to the ability of representations to generalize or not to novel stimuli and tasks. They first look at artificially created latent representations and the high-level/latent representations of feedforward ANNs (generally the representations of the last layer of a feed-forward network before the output layer). Geometry is quantified by the ability of the classifier to generalize to novel instances of the stimulus set, and by quantifying whether a linear regression on the hidden representations generalizes to a novel stimuli. Using a number of different types of networks they observe that the

representations become more abstract/linear as the number of tasks increases. Thus one of the predictions made, is that the abstractness of neural representations in animals should increase with the number of tasks the animal is trained on.

We thank the reviewer for their helpful comments on our manuscript.

1.2 Major comments

1.2.1

The geometry of representations is an important, and increasingly studied, problem in neuroscience and machine learning. In ML it is well established that effective word embeddings are abstract and as a result of their linear properties allow for “word algebra”. The standard example being that the representation of KING – MAN + WOMAN = QUEEN; and that these embeddings benefit from large datasets (not that words correspond to directly to number of tasks). In the ML field the findings are probably expected, although some of the methodologies may be novel. But it may be helpful to determine if such representational algebra applies to the stimulus classes studied here.

We agree that representational algebras like the example given here are an important facet of abstract representations. In the new version of the manuscript we show within our framework that the representational algebra can be applied to simple image representations, see the figure fig. S5. We discuss these results in the main text,

(line 487) Finally, we also use the image setting to investigate one important property of abstract representations that is not captured by the standard multi-tasking model: compositionality of representations. In machine learning, abstract representations are desirable primarily because they allow representations to be composed to produce output representations with predictable features[1]. To investigate this in our setting, we train a multi-tasking model model on the shape image dataset, where the multi-tasking model must perform binary tasks as before, but is also tasked with reconstructing the original image input from the representation layer as well (see *The multi-tasking model can be used as an abstract, generative model* in *Supplement* for the details of the model). Then, we learn a vector representation of shape scale from two of the three shapes included in the dataset (fig. S5d). Next, we take the representation for the third shape at a starting scale and use the learned vector to produce shape examples with increased and decreased scale (fig. S5e). Thus, not only does the representation of scale generalize across the different shapes, but this property can be used to generate images with a desired scale in a compositional way.

1.2.2

I found the paper rather difficult to follow and I suspect other readers will as well. Part of the confusion arises because of the multiple meanings of “model” which can refer to the feedforward network or the decoders used to analyze the representations in the feedforward network. Whether the network or the decoders are being trained on multiple tasks also is a source of confusion. Additionally, some panels are called out of order (e.g., 2a is called before 1f), and I don’t think Figs. S1 S5 S6 S7 S8 are mentioned in the main text. In the early description of the Results, the explanation of the methods is not very

clear (e.g. clarify which latent representations the two abstraction metrics are performed on). Finally, too much critical information needed to understand the early figures is referred to the Methods, making it impossible to understand the Results without first reading the Methods very carefully. For example, in the first paragraph of Section 2.2 there are three calls to the Methods and Supplements, making it hard to understand the key results of Fig 2 without a lot of back and forth.

We agree that the previous version of our manuscript was unclear on several points and have taken several different steps to remedy this. First, we have added a new figure that serves to, first, introduce our geometric notion of abstraction and, second, orient the reader as to the organization of and interaction between the various different "models." The figure is reproduced here for convenience (fig. RR1). Through the figure, we introduce a new color code that has propagated through all of our other main figures, where green represents the input model, blue represents the multi-tasking model, and pink represents the abstraction metrics.

We have also added additional explanatory text,

(line 105) Here, we study how abstract representations like the ones in our example emerge for stimuli described by D continuous latent variables in a feedforward neural network. The latent variables themselves are already abstract. So, we begin by constructing a nonlinear and non-abstract representation of the latent variables to use as our input going forward (fig. 1c), which we refer to as the standard input. Then, we introduce the multi-tasking model, which receives these non-abstract representations of the latent variables as input (fig. 1d, left) and is then trained to perform P random binary classification tasks on the latent variables (fig. 1d, right). Finally, after the multi-tasking model is fully trained, we quantify the level of abstraction developed in its representation layer using two abstraction metrics (fig. 1e).

as well as

(line 145) Importantly, each of these three components of our framework are trained in sequence to each other: The input model (fig. 1c) is trained first and then frozen. The input model is used to generate the training data for the multi-tasking model (fig. 1d), which is trained second. Then, finally, we use our abstraction metrics (fig. 1e) to quantify the level of abstraction present in the representation layer of the trained multi-tasking model (and in the trained standard input, as in fig. 2).

We have also rewritten several parts of the results to make them both more clear and self-contained. For instance,

(line 305) The multi-tasking model is trained to simultaneously produce output for P different random tasks. Importantly, the standard input used in this section already has high classification performance for random hyperplane tasks on the latent variables (fig. 2f, left), due to its high embedding dimensionality[2]. So, one possibility is that the representation layer in the multi-tasking model would retain the same, non-abstract structure. However, our results in the previous section and experiments with multi-tasking models that are trained with layers that all have the same width as the input (see *The effect of constant layer widths on abstraction* in *Supplement* and fig. S11) show that this is not the case. Instead, the multi-tasking model develops robustly abstract representations (fig. 3e, f).

Finally, we have also ensured that all of our supplemental figures are now referenced in the main text.

Figure RR1: The abstraction metrics and input representations. **a** Two example classification tasks. (left) A classification learned between red and blue berries of one shape should generalize to other shapes. (right) A classification between red berries of two different shapes should generalize to blue berries of different shapes. **b** Examples of linear, abstract (left) and nonlinear, non-abstract (right) representations of the four example berries. **c** Schematic of the input model. **d** Schematic of the multi-tasking model. **e** Schematic of our two abstraction metrics, the classifier generalization metric (left) and the regression generalization metric (right).

1.2.3

Here the latent representations were high-dimensional and non-sparse. Neural representations sometimes appear to be quite sparse (e.g., in the ventral visual stream). Do the current results hold for more sparse representations?

We were unsure whether the reviewer meant that the standard input was high-dimensional and non-sparse (though this representation may not really be "latent") or that the multi-tasking model representations are high-dimensional and non-sparse. To clarify, what we find is that the standard input is both high-dimensional and sparse (we had not emphasized the sparsity before, but now we do both in the main text and in fig. 2 where we show standard input receptive fields). We also find that the representations in the multi-tasking model are relatively low-dimensional and non-sparse (though we had not quantified this before either).

In the multi-tasking model representations, we agree that this is an interesting potential difference between the abstract representations developed by the multi-tasking model and representations observed in the brain. To explore it in more detail, we trained the multi-tasking model with several different levels of L_1 and L_2 regularization applied to activity in the representation layer. Then, we used a common measure of sparseness (introduced in the text) to quantify the level of sparsity developed by both the original multi-tasking model and the multi-tasking models trained with regularization. We found that – in agreement with the reviewer’s intuition – the activity in the

original multi-tasking model is not particularly sparse. However, the multi-tasking models trained with regularization develop significantly more sparse representations with only a mild decrease in abstraction. These results are described in more detail in *The effect of activity regularization on abstraction* in *Supplement* and see fig. S9.

We also added some discussion of this kind of both sparse and abstract representation to the Discussion – and linked it to representations that have been observed in the brain (such as some representations in the inferotemporal cortex) as well as representations studied in machine learning,

(line 541) Representations in the brain are often observed to be sparse[3]. Here, while the standard input and RF input (fig. S4) that we explore are highly sparse, the abstract representations that the multi-tasking model develops are not necessarily sparse. Indeed, when we characterize the sparseness of representations in the multi-tasking model, we find that they are substantially less sparse than the inputs (fig. S9a, c, left). To explore this apparent inconsistency, we apply regularization to the activity in the representation layer of the multi-tasking model. In models trained with weak L_1 and L_2 regularization, we find only a small decrease in the classification and regression generalization performance (fig. S9b, d) along with a striking increase in the average sparseness across the population (though it remains less sparse than the input, fig. S9a, c). Thus, sparseness and abstract representations can coexist in the multi-tasking model. Further, the representation of facial features in the brain is thought to share this property: In the whole population of inferotemporal cortex neurons, face selectivity is relatively rare – and so the representation is sparse[4] (though face cells are also concentrated in particular anatomical subdivisions of the inferotemporal cortex[5]). However, within face-responsive neurons, the code is almost linear in facial features[6] and is abstract[7, 8]. We can view this as two hierarchical codes. The outer code is a sparse representation of object identity (e.g., face or hand). The inner code is a dense, abstract code for the features of that object (e.g., a happy or sad expression). This may be a general strategy for object representations in the primate brain[9]. Further, this particular kind of sparse representation has been explored in machine learning[10–12] and is thought to be essential for flexible and intelligent behavior[13].

1.2.4

Are the results dependent on the “dimensionality” (number of units in the representation) of the representations? It would seem that the higher the dimensionality the larger the number of tasks would have to be for the linearity to arise?

We have included additional experiments that explore this point. We were unsure whether the reference to “dimensionality” (number of units in the representation) referred to the input representation or to the multi-tasking model representation layer (another ambiguity, related to the earlier comment about the ambiguity of model, which we have worked to address in the new version of the manuscript). In any case, we felt both were interesting to address. We address the first in the main text with,

(line 263) Next, we test how robust these abstract representations are to increases in the embedding dimensionality of the input... First, we show that this finding is almost unchanged given standard input models that produce higher-dimensional input (fig. S10 and see *The effect of increased input dimensionality on abstraction* in *Supplement*).

and the second,

(line 306) Importantly, the standard input used in this section already has high classification performance for random hyperplane tasks on the latent variables (fig. 2f, left), due to its high embedding dimensionality[2]. So, one possibility is that the representation layer in the multi-tasking model would retain the same, non-abstract structure. However, our results in the previous section and experiments with multi-tasking models that are trained with layers that all have the same width as the input (see *The effect of constant layer widths on abstraction* in *Supplement* and fig. S11) show that this is not the case. Instead, the multi-tasking model develops robustly abstract representations (fig. 3e, f).

1.2.5

It is not clear if the clear linear/abstract representations are in part dependent on the assumption used, including the linear decoding and the regularization. That is, is it necessarily the case that if you are using a linear decoder you inevitably get linear representations as the number of tasks increases? Is there a mathematical theorem that states that since one is reducing the error during training using standard loss functions, demonstrates that the linearity of the representations must increase as one trains more and more tasks using linear readouts?

First, we want to clarify that when we say linear decoder, we mean a binary classifier that uses a linear hyperplane to separate the representation space into two distinct categories. We are not training the multi-tasking model to read out continuous variables that are linearly related to the underlying latent variables, we have also clarified this in the text where we first introduce our tasks

(line 109) ... which receives these non-abstract representations of the latent variables as input (fig. 1d, left) and is then trained to perform P random binary classification tasks on the latent variables (fig. 1d, right)

and

(line 226) Finally, we also visualize the projection of the representation layer from each of multi-tasking models onto one of the task outputs (fig. 3d). We see that for one and two tasks (fig. 3d, top and middle), the task output value is strongly separated and bimodal.

and we now visualize the representation that is learned when the multi-tasking model is trained on a single task (or two tasks) in fig. 3, which shows their nonlinearity (particularly bimodality, as mentioned above).

We also want to stress that we relax the assumption that these tasks have linear hyperplanes in fig. 5 and fig. S2, and show qualitatively similar results (and see fig. 4, where we show that even mixtures of nonlinear and linear classification tasks are sufficient to produce abstract representations). In the theory, we show that training a feedforward network to perform multiple classification tasks with linear boundaries using standard backpropagation and no regularization will increase the strength of a component of the representation with dimensionality approximately the same as either the number of tasks or the number of latent variables, whichever is lower. We show that this approximation is closer for larger numbers of latent variables. In the text, we discuss this finding,

(line 316) This means that, given application of backpropagation, the representation layer will be dominated by a $\min(P, D)$ -dimensional representation of the latent variables. Since this representation must also be able to satisfy the P tasks, it will at least have high classifier generalization performance and may even have high regression generalization performance (see *Four possibilities for representations in the multi-tasking model* in *Supplement* for more discussion of alternative representations). While the multi-tasking model used in the rest of the paper has a sigmoid output nonlinearity, the intuition developed in this simplified case still applies.

In the supplement, we also discuss other possible kinds of representation that has this dimensionality constraint,

(line 999) We consider four distinct kinds of representations that could support the simultaneous performance of P classification tasks as formalized in the multi-tasking model. First, the high dimensional standard input could be preserved, or only weakly tuned – in particular, recall that high classification performance is already achieved on the standard input for random tasks (fig. 1f, left "standard") even though it is not abstract (fig. 1f, left "gen"). Second, the representation could split along P separate dimensions of population activity, where each dimension corresponds to one of the P distinct tasks (fig. S1b, left). Third, the representation could consist only of an approximately D -dimensional sphere (or circle, in two dimensions), which exploits the correlation structure in the P different tasks (that is, when $P > D$, the outcomes from some pairs of tasks are necessarily correlated with each other; fig. S1b, middle). This second type of representation would have high classifier generalization performance but low regression generalization performance: That is, it is partially abstract in that it would recover the angular structure of the latent variables (as necessary for the P classification tasks), but not their magnitude (as this information is not necessary to solve the P tasks). Fourth, a fully abstract representation of the latent variables could be recovered. That is, the representation could recover both the angular structure of the latent variables, as in the second possibility, and their magnitude (fig. S1b, right). This would occur only if the multi-tasking model does not discard information about the stimuli that is not necessary for satisfying the tasks, but which is also not explicitly trained to discard. Surprisingly, as we will see, this fourth form of representation is most common in our trained networks, even for more disordered tasks than we have described so far.

and have a schematic visualization in fig. S1.

Finally, we also have now included a supplemental figure showing that the inclusion of an L1 or L2 penalty on activity in the representation layer of the multi-tasking model produces only a minor decrease in the level of abstraction of the representation, while significantly increasing the sparsity of the representation (fig. S9).

1.2.6

In Fig 6c why is it that the fraction of tasks learned start low go up and come down? Is this related to abstract representations? Provide the n's and SEM for the curve in the panel.

We thank the reviewer for catching this issue in the plot. We reworked the plot to average across

more runs, and have now included the n ($n = 10$) and SEM. In short, averaging over more runs changes the profile of the curve, so that it now increases in the number of tasks.

1.2.7

The term “dimensionality” is a bit confusing. Generally, there are several types of dimensionality: the intrinsic dimensionality—the intrinsic geometric properties of a given manifold or latent space, such as a 2D sphere; the extrinsic dimensionality of a manifold dependent on how the manifold is embedded in the ambient space; and the dimensionality of the ambient space itself. The author might want to clarify such a definition whenever it is referred.

We have worked to correct this ambiguity. In particular, we now explicitly use “embedding dimensionality” rather than simply “dimensionality” throughout the paper for discussing the embedding dimensionality of representations.

1.2.8

Reasonably so, the authors focus on the last layer for the representation since it is closest to the output. The reader might also want to know how such representations evolve through different layers, in other words, is there a progressive development of abstract representations through layers and is it also dependent on the number of tasks.

We have now quantified this (fig. S12) and included a reference to the results in the main text,

(line 197) In all of our analyses, we focus on the representations of the stimuli that are developed in the layer preceding the task output layer, which we refer to as the representation layer (but see fig. S12 and *Abstraction emerges even in earlier layers of the multi-tasking model* in *Supplement* for an analysis of the other layers).

and, in the supplement,

(line 1202) We have explored several different numbers of hidden layers for the multi-tasking model and found high levels of abstraction for all of them. So, we asked whether abstract representations also develop in earlier layers of the multi-tasking model or only close to the output, in the representation layer we have studied so far. We find that in both multi-tasking models with the standard width and with uniform widths, abstract representations develop after the first hidden layer, after only one nonlinear step (fig. S12a, b). However, the classifier generalization performance slightly increases in the second hidden layer (fig. S12a, b). The representation layer has the same classification and regression generalization performance as the prior hidden layer, which is expected because it is a linear transform of the second hidden layer.

This result is somewhat surprising, but follows the intuition: Representations with high embedding dimensionality (such as the standard input) can be directly transformed to abstract representations when supplied with appropriate training information (here, the outcomes of several classification tasks).

1.3 Minor comments

1.3.1

Sections 2.5 and 2.6 discuss the same figure, but called the panels in a confused way: the results were referred in 2.5 before it was explained how it was generated up to section 2.6.

We have now reworked this figure as well as the order in which it is discussed in the text. We agree that our previous discussion of it was too complicated in order. We begin by introducing the Gaussian process approach, then the results from the Gaussian process inputs, then the results from the outputs – mirroring the earlier progression of the paper. The new introduction of the approach reads,

(line 370) To understand the constraints on the multi-tasking model’s ability to learn abstract representations from non-abstract input, we introduce both a new input model (fig. 5a, b, c) and a new kind of nonlinear task (fig. 5a, d, e). We control the length scale of correlations in both the input model and the tasks. Then, we quantify the classification and regression generalization performance as we vary both length scales simultaneously (fig. 5f, g).

1.3.2

Fig 3, panels b and d are confusing, it looks like the right panels are part of the same panel. Having separate y axis labels might help.

We added more panel labels to this figure to increase the clarity.

1.3.3

There are a number of typos, which are a bit hard to list in the absence of line numbers in the manuscript. For example:

Pg. 7 “an multi-taking model” Pg. 15 “rapid objection recognition” Pg. M2 “we we”

We have fixed these typos (as well as several others) and added in line numbers.

2 Review 2

2.1 Summary

The authors show that when neural networks are trained to perform multiple tasks that rely on linearly classifying latent variables, the representations that emerge are linear embeddings of the latent variables. These linear embeddings are an efficient substrate for learning and, in particular, allow a classifier or regression model trained on one region of latent space to generalize to other regions of latent space.

I thought this was an interesting, well-executed study that seeks to address an important question. My major concerns were about the way in which the task structure naturally imposed the desired representation and thus whether the movement from task structure to representation really captures what we mean by abstraction. I have elaborated this and other concerns below:

We thank the reviewer for their helpful comments on our work. We hope that our additional work to clarify and further visualize our results helps to assuage the reviewer’s concerns about the task structure directly imposing the desired representations (though of course we agree that the task structure is crucially important to our results).

2.2 Major comments

2.2.1

1) The study relies quite heavily on linearity—the notion of abstract representations is that they are linearly represented in neural population activity space and the tasks that the network is trained on correspond to linear separation in latent variable state. Given that the final transformation from representation layer to readout layer is near linear, as far as I understand, the theoretical argument is that if one trains a network to read out (at least) D linear functions of D variables using linear projections, then the representation must necessarily be linear in the D variables.

The reviewer has several comments on this concern which we address in detail below. However, we want to begin by clarifying that the network is not learning to read out D linear functions of D variables. While the network is learning binary classification tasks with linear category boundaries, the task itself is a nonlinear function of the D latent variables – and, as we now show, learning 1 of these functions does not induce a linear representation of the underlying latent variable, instead it induces a bimodal representation (fig. 3d). We also want to stress that we relax the assumption that these tasks have linear hyperplanes in fig. 5 and fig. S2, and show qualitatively similar results (and see fig. 4, where we show that even mixtures of nonlinear and linear classification tasks are sufficient to produce abstract representations). We also discuss some other specific concerns around this in later comments.

2.2.2

I think the first use (abstract representation=linear) is an interesting formulation of abstractness but could be unpacked a bit more and contrasted to other notions of “abstract” (besides the beta-VAE). For example, how does it relate to notion of abstraction

as extracting modular, compositional structure from the world? I’m guessing that the implicit assumption is that the linearly separable factors are the modules that can be flexibly composed but would be nice to have this explicit.

A representation based on a modular structure in which each module represents a different subset of latent variables is certainly abstract in our sense, it allows for generalization and compositionality – and we now make this connection more explicit in the main text,

(line 600) These single neuron representations of single latent variables lead to distinct modules within the neural population, one module for each latent variable. This kind of representation would also be abstract under our metrics, and can be viewed as a special case in which the axes of neural population space are aligned with the latent variables. Our abstraction metrics, however, do not require this alignment. They depend on the geometry of the representations at the population level and this geometry is unaffected by whether single neuron activity corresponds to a single latent variable, or to a linear mixture (i.e., a weighted sum) of all the latent variables. Given the extensive linear and non-linear mixing observed already in the brain[2, 6, 14, 15], we believe that this flexibility is an advantage of our framework for detecting and quantifying the abstractness of neural representations.

However, we also want to stress that a modular representation is not the only kind of representation that has these properties. The latent variables can be disentangled even when there is no modularity: a rotated modular representation would not preserve modularity but it would have the same representational geometry and hence the same computational properties. In particular, it would also allow for generalization and compositionality. In the previous version of the manuscript we focused only generalization, but we agree with the reviewer that compositionality is an interesting and important property and hence we have now included a discussion of results where we directly show that the abstract representations developed in the multi-tasking model can be used in a compositional manner,

(line 487) Finally, we also use the image setting to investigate one important property of abstract representations that is not captured by the standard multi-tasking model: compositionality of representations. In machine learning, abstract representations are desirable primarily because they allow representations to be composed to produce output representations with predictable features[1]. To investigate this in our setting, we train a multi-tasking model on the shape image dataset, where the multi-tasking model must perform binary tasks as before, but is also tasked with reconstructing the original image input from the representation layer as well (see *The multi-tasking model can be used as an abstract, generative model* in *Supplement* for the details of the model). Then, we learn a vector representation of shape scale from two of the three shapes included in the dataset (fig. S5d). Next, we take the representation for the third shape at a starting scale and use the learned vector to produce shape examples with increased and decreased scale (fig. S5e). Thus, not only does the representation of scale generalize across the different shapes, but this property can be used to generate images with a desired scale in a compositional way.

2.2.3

2) I have concerns with the second use of linearity from the point above—the close match between the desired representations and the trained tasks is a strong assumption. In essence, it seems like each task causes the network to learn a linear representation along one dimension in latent space and D such independent tasks force abstraction (or linearity) along D dimensions (though I appreciate the authors’ point that this is not the only solution that can be conceived of).

We have added clarification to the main text as well as included some visualizations in the figure to show that training the network to perform a single task does not induce a linear representation along that task dimension (fig. 3),

(line 226) Finally, we also visualize the projection of the representation layer from each of multi-tasking models onto one of the task outputs (fig. 3d). We see that for one and two tasks (fig. 3d, top and middle), the task output value is strongly separated and bimodal.

Most importantly, as the reviewer mentions below, we have explored multiple relaxations of the linear category boundary (contextual boundaries, curved boundaries, mixtures of grid and linear boundaries). In all cases, we found qualitatively similar results (fig. 4, fig. 5, fig. S2).

2.2.4

This way of abstracting seems quite different from a more usual notion where each single or small subset of tasks can be performed without abstraction but an organism or algorithm learns to construct abstract representations because they are useful and efficient across distinct tasks. It also seems distinct from learning modular computations, representations or dynamics that can be reused across tasks (e.g., the Yang 2019 paper). If this is a fair characterization, I think the paper should at minimum include some more discussion of the role this match between desired representations and task structure plays.

We agree that there are a few different ways in which the term abstraction has been used previously in the literature. We now discuss the second form of abstraction mentioned by the reviewer (modular representations and operations) in more detail in the Discussion. As quoted above, we discuss the connection between modularity and our abstraction metrics: in short, modular representations are abstract under our framework and the representations that we find in the multi-tasking model have the desirable properties of modularity. We also discuss related work on modular operations or computations in recurrent neural networks,

(line 583) ... there is a growing literature on the ability of network dynamics to implement abstract operations. In particular, recent work has shown that training recurrent neural networks to perform multiple dynamic tasks leads to shared implementations of common task operations (such as storing information across a delay period)[16–18]. As a result, novel tasks can be quickly acquired through the combination of these learned abstract operations[17]. This is an important form of abstraction that differs from the abstract representations we have studied here. We believe that the two forms can work in tandem: Abstract representations (in our sense) may be important for the abstract operations to be robust to irrelevant changes in context. However, our work suggests that these abstract representations may emerge naturally from the multi-task training that these networks already undergo. We believe that further work can fruitfully

combine these two lines of research.

As for the first kind of abstraction, we argue that this is what we are investigating here. In particular, our findings show that learning some tasks both increases the efficiency of learning a novel, related task as well as the ability to learn that task from one context and generalize correctly to a novel context. Notably, this is not true when we train our networks to perform only one or two tasks. In fact, when we train our network to perform only one task, it learns a specialized, bimodal representation that cannot usefully generalize to a novel task (fig. 3). We also note that, in most cases, the novel tasks can be successfully learned on the input provided to the multi-tasking model. However, when learned on the input, they do not generalize well across contexts. The ability to generalize across context relies on the information provided by the other tasks learned during training.

2.2.5

Note that the existing simulations of contextual and partial information task partitions and the more complex GP task boundaries go some way to addressing my concerns, and make me wonder if there's some picture in which multiple locally linear task boundaries are averaged or stitched together across multiple tasks to get global linearity. If true I think this would be very interesting and could be highlighted as a way in which multiple tasks are fundamentally important.

We agree that this local averaging idea is a likely way that the learning of multiple tasks helps to provide generalization, especially in the case of the contextual tasks which, by construction, only provide short length scale information about the underlying latent variables. We now discuss this idea more in the text,

(line 285) In the case of the contextual tasks, the latent variable information provided by the tasks is necessarily partial. To develop abstract representation even in this case, the multi-tasking model must combine information from multiple different contextual tasks.

In addition, we have done some preliminary work to explore this idea. In particular, we trained multi-tasking models to perform different numbers of contextual tasks and then visualized the response fields of units in the representation layer of the trained model (fig. RR2). Our findings are somewhat inconclusive. While the visualization shows that networks trained to perform one or only a handful of tasks develop stereotyped response fields and networks trained to perform more tasks develop a more diverse set of responses, we have struggled to conclusively relate this to the different contextual task boundaries that the network learns. As a consequence, we have chosen to leave further investigation of this case for future work rather than to introduce a major new conceptual approach into this paper.

2.2.6

3) Related to the above, the novel tasks are set up so that they can be learned as a one layer linear readout from the D dimensional representation so I would expect the performance to be identical to learning directly from the latent variables (as the authors show). Do the results hold for a more complex novel task, which for example might require tuning weights in earlier layers? Maybe the GP task boundaries from later in the study could be used?

Figure RR2: Receptive fields for learned contextual tasks. Responses in the representation layer trained to perform different numbers of tasks (columns) along a two-dimensional slice of latent variable space (response magnitude is given by color). Notice the lack of diversity in networks trained to perform few tasks relative to many tasks.

We explored this idea by running a learning efficiency analysis similar to the one we had already used in the paper (fig. 3) where the novel task that is being learned is no longer a random linear task, but instead a random Gaussian process task. We show that this also leads to reliable learning of and good generalization on the novel task with relatively few samples,

(line 439) Finally, instead of training the multi-tasking model using random Gaussian process tasks, we explored whether or not the network representations could be used to efficiently learn and generalize on novel random Gaussian process tasks instead of the linear tasks that we have been using to quantify abstraction so far. We found that, across several different length scales, both the sample efficiency and generalization performance on the novel, curved task were close that of learning directly from the latent variables (fig. S13 and see *Novel random Gaussian process task learning in Supplement* for more detail; this mirrors the efficiency and generalization performance of learning a novel linear classification task, fig. 3f). Thus, the abstraction representations learned by the multi-tasking model facilitate efficient learning and generalization even when the novel task is not linear.

2.2.7

4) Also related to the linearity constraint, the authors often say high dimensionality when they mean nonlinearity / high embedding dimension. E.g., “high-dimensional representation often do not permit generalization” might be true, but the claim here is about low-dimensional but nonlinear representations. The authors are of course aware of the distinction but it should be made clear which is meant throughout.

We agree that there is some ambiguity here. We mean to say that high-embedding dimension representations of lower-dimensional latent variables (low intrinsic dimensionality) often do not permit generalization. We interpret the reviewer as wanting us to clarify that we are not making any claim about whether high-dimensional latent variables (high intrinsic dimensionality) permit generalization. We have included clarification that it is specifically nonlinear mixing of different latent variables that reduces the ability to generalize. In the main text,

(line 31) Neural representations of sensory and cognitive variables are often nonlinearly mixed together. As a result, these representations have high embedding dimension[2, 19, 20]. While this kind of nonlinear dimensionality expansion allows flexible learning of

new behaviors[19] and provides metabolically efficient and reliable representations[21], the resulting representation often does not permit generalization across contexts or stimuli[15, 19].

We have also clarified throughout the text that we are referring to embedding dimensionality when we discuss the dimensionality of representations.

2.2.8

5) In the sensitivity analysis, does the shrinking size of deeper layers play an important role? Would the network learn the low-dimensional representation (as opposed to just copying the standard input) if deeper layers were the same size? It looks from Fig. S7 like doubling the size of just the representation layer doesn't have an effect but is a bottleneck at some point important?

We agree that this is an interesting question, especially as this kind of bottleneck can be important to other methods of producing disentangled representations. We investigated this by training a multi-tasking model that has layers that are all the same width as the input (except the output layer, which is still the same size as the number of tasks). The increase in width throughout had almost no effect on the abstraction of the representations,

(line 308) So, one possibility is that the representation layer in the multi-tasking model would retain the same, non-abstract structure. However, our results in the previous section and experiments with multi-tasking models that are trained with layers that all have the same width as the input (see *The effect of constant layer widths on abstraction in Supplement* and fig. S11) show that this is not the case. Instead, the multi-tasking model develops robustly abstract representations (fig. 3e, f).

This is included as fig. S11. The result follows from our theory: The low-dimensionality in the representation is inherited from the required output (due to the correlation between tasks), not from a bottleneck enforced by a reduced number of units in the representation layer. While the reviewer is correct that one way to achieve high performance on all the tasks would be to simply mimic the input, our theory shows that the training process increases the strength of a lower-dimensional component of the representation that is aligned with the latent variables. It is necessary, of course, that the layers of the multi-tasking model all have more units than the dimensionality of the latent variables ($D = 5$ in the text), but we have found no indication of any constraints other than that.

2.2.9

6) Sections 2.5 and 2.6 were confusing and took me several reads (though the analyses themselves are really nice and I appreciate the authors' thoroughness). I think this is entirely because of how the figure is organized and the text itself is relatively clear (though 2.6 could be clarified a bit more). Specifically, as far as I can tell, Section 2.5 is about using curved boundaries in output task space rather than linear ones, and Section 2.6 is about using a different parameterization of the input (i.e., GP tuning curves) but the corresponding figure is interleaved between sections and in particular Fig. 4a goes with section 2.6. I would suggest just splitting into two smaller figures that go with the appropriate section, or if the authors must keep a big figure, to put anything related to Section 2.5 before 2.6.

We agree that our description of this figure was hard to follow due to its order. We have reorganized both this figure and the accompanying text. In particular, we now introduce the whole Gaussian process approach before discussing the input and output components in more detail separately. We believe this helps to alleviate the confusion.

2.2.10

7) The lower performance for the GP input and the receptive field input seems like an important issue, given that GP tuning curves and localized receptive fields are closer to biology than the autoencoder. Up until this section, my read was that the autoencoder used to generate the standard input just generated some highly nonlinear transformation of the input that was then unlearned by the multitasking model. But these results suggest that there’s something special about the autoencoder input. Can performance like the autoencoder input be recovered with these more natural input structures and for what depth of network or objective function does this happen? If not there might be an interesting scientific point to be made here about the need for a two-stage sensory/cognitive model (with a separate loss function for the sensory representation), especially given the results in Section 2.7 that training the multitasking model on top of an object recognition model yields abstraction.

We have addressed this in two ways. First, we reworked our approach to the receptive field input – and found that, while regression generalization performance remains lower, it is now significantly above chance. In fact, we found that the largest difference in performance was due to the different numbers of latent variables used originally for RFs ($D = 2$) and for the standard input ($D = 5$). The standard input also shows lower performance for lower dimensional latent variables (fig. S4), and we discuss potential reasons why this is in *The dependence of learned abstract representations on latent variable dimensionality* in *Supplement*.

Second, we agree that a two-stage transformation may be important, particularly in the case of the random Gaussian process inputs. We discuss this in the text here,

(line 560) While we find fully abstract representations for the standard input (fig. 3), receptive field inputs (fig. S4), and image inputs (fig. 6), we do not find fully abstract representations for low length scale random Gaussian process inputs (fig. 5f, g). The low length scale random Gaussian process input differs from all other input types in one important way: Both linear decoders and regressions perform relatively poorly even when trained and tested on the whole stimulus space (fig. 5d). Thus, this initial linear separability may be a prerequisite for the multi-tasking model to produce abstract representations. Further, it suggests that a crucial step may be an initial dimensionality expansion, that produces this separability, before the dimensionality of the representation is collapsed again into an abstract form. Future work will investigate incorporating this into the multi-tasking model through regularization of the first layer.

We also note that there is also a difference in required output in many cases we investigated with the random Gaussian process inputs, and we fail to find abstract representations in cases where both the input is highly tangled (low length scales) or the output is highly nonlinear (low length scales). This latter case is similar to the grid tasks from before – and thus we would not expect abstract representations in this setting. However, when the output length scale is long, it is equivalent to the linear tasks used elsewhere in the manuscript. Here, we find abstract representations over a

wider range of input length scales, as expected.

2.3 Minor comments

2.3.1

8) The paper gets to geometry, nonlinearity of representation, high embedding dimension, etc. quite quickly (e.g., "Neural representations of sensory and cognitive variables are often highly nonlinear and have high embedding dimension"). For accessibility to non-theorist audiences this should be unpacked a bit, even if only a paragraph quickly summarizing the geometric view of neural coding.

We have included additional introduction of the geometric view of neural coding as well as unpacked how we introduce the specific geometries that we are interested in here. The section now reads,

(line 28) The representational geometry of sensory and cognitive variables in a population of neurons provides insight into the computations that the representation may and may not facilitate[22–24]. We hypothesize that the ability to generalize described above is tied to this representational geometry. For instance, neural representations of sensory and cognitive variables are often nonlinearly mixed together. As a result, these representations have high embedding dimension[2, 19, 20]. While this kind of nonlinear dimensionality expansion allows flexible learning of new behaviors[19] and provides metabolically efficient and reliable representations[21], the resulting representation often does not permit generalization across contexts or stimuli[15, 19]. Alternatively, factorized, or even linear, representations of the relevant sensory or cognitive variables (i.e., representations that have no nonlinear mixing) often permit this generalization.

2.3.2

10) Pg. 4 "Feedback onto representation layer" I'm guessing feedback through learning since it's a feedforward network? If so "feedback" is confusing.

We agree that this was confusing, we have changed the wording here and it now reads,

(line 308) So, one possibility is that the representation layer in the multi-tasking model would retain the same, non-abstract structure. However, our results in the previous section and experiments with multi-tasking models that are trained with layers that all have the same width as the input (see *The effect of constant layer widths on abstraction in Supplement* and fig. S11) show that this is not the case. Instead, the multi-tasking model develops robustly abstract representations (fig. 3e, f).

2.3.3

11) What are gray lines in Fig. 2b and c?

We have clarified what the grey lines are in both the main text and the figure caption. From the main text,

(line 252) We compare this performance to a lower bound (fig. 3f, dark grey), from when the task is learned from the standard input representation; as well as an upper bound

(fig. 3f, dark grey), from when the task is learned directly from the latent variables. The performance of the multi-tasking model nearly saturates this upper bound (fig. 3f, left).

2.3.4

12) Equation at top of Pg. 6: I guess x is input, $r(x)$ is value in representation layer and W is connections in last layer. But this should all be defined and argument unpacked slightly so argument can be read without looking ahead to Methods.

The reviewer was indeed correct in their inference, but we apologize for not including the necessary definitions. We have now included them in the main text,

(line 313) where x is a stimulus sampled from our latent variable distribution, W are the weights connecting the representation layer to the task outputs, $r(x)$ is the activity corresponding to stimulus x in the representation layer, and A is a $P \times D$ matrix of randomly selected task vectors (i.e., the vectors that define the binary classification hyperplane).

2.3.5

13) The linear representation is necessarily dense, with most neurons active for most inputs. How do the authors see this as comparing to sparse activation in neural populations?

We have addressed this in two ways. First, by quantifying how our results change when the multi-tasking model representations are encouraged to be sparse by L_1 and L_2 activity regularization,

(line 543) Indeed, when we characterize the sparseness of representations in the multi-tasking model, we find that they are substantially less sparse than the inputs (fig. S9a, c, left). To explore this apparent inconsistency, we apply regularization to the activity in the representation layer of the multi-tasking model. In models trained with weak L_1 and L_2 regularization, we find only a small decrease in the classification and regression generalization performance (fig. S9b, d) along with a striking increase in the average sparseness across the population (though it remains less sparse than the input, fig. S9a, c). Thus, sparseness and abstract representations can coexist in the multi-tasking model.

Second, we also discuss these kinds of representations more generally, and how they fit into our existing understanding of representations in the brain (and in machine learning systems),

(line 550) Further, the representation of facial features in the brain is thought to share this property: In the whole population of inferotemporal cortex neurons, face selectivity is relatively rare – and so the representation is sparse[4] (though face cells are also concentrated in particular anatomical subdivisions of the inferotemporal cortex[5]). However, within face-responsive neurons, the code is almost linear in facial features[6] and is abstract[7, 8]. We can view this as two hierarchical codes. The outer code is a sparse representation of object identity (e.g., face or hand). The inner code is a dense, abstract code for the features of that object (e.g., a happy or sad expression). This may be a general strategy for object representations in the primate brain[9]. Further, this particular kind of sparse representation has been explored in machine learning[10–12] and is thought to be essential for flexible and intelligent behavior[13].

2.3.6

14) Other recent work has found that low-dimensional linear (i.e., abstract) structure emerges in network that are trained to perform predictive learning (Recanatesi et al. 2021). Is there a relationship between these frameworks? If so making that explicit would be useful to the field.

We agree that this predictive learning framework may be related to our multi-tasking framework. We have included an explicit discussion of that potential connection,

(line 577) Interestingly, recent work has shown that neural networks trained to predict the result of a chosen action develop low-dimensional, potentially abstract representations of the latent space underlying the observations[25]. This form of prediction could be viewed as a multi-tasking problem similar to the one we studied here – and could indicate that abstract representations may emerge naturally from predicting the sensory consequences of our actions, without explicit feedback.

2.3.7

15) Note repeated sentence on Pg. S-1 and S-4 “However, we believe the contrast is still informative. . .”

We have removed the repeated sentence.

Reviewers' Comments:

Reviewer #1:

Remarks to the Author:

I think the authors have done a good job improving the manuscript and explaining the models. The new figure panels were very helpful.

Figs 6 and 7 have the wrong panel labels, and the text above the colored horizontal lines is garbled.

Overall I think the results provide valuable insights to the potential importance of linear representations and how they may emerge in the brain.

Reviewer #2:

Remarks to the Author:

I thank the authors for their careful response to the review, and my concerns have been addressed in the revision.

Response to the reviewers

We appreciate the attention given to our manuscript by both reviewers. We have corrected the minor errors in the captions for Figures 6 and 7 pointed out by reviewer 1.

1 Review 1

I think the authors have done a good job improving the manuscript and explaining the models. The new figure panels were very helpful.

Figs 6 and 7 have the wrong panel labels, and the text above the colored horizontal lines is garbled.

Overall I think the results provide valuable insights to the potential importance of linear representations and how they may emerge in the brain.

We are glad that we were able to address the reviewer's concerns and we thank them for their continued attention to the manuscript. We have corrected the errors in Figures 6 and 7.

2 Review 2

I thank the authors for their careful response to the review, and my concerns have been addressed in the revision.

We are glad we have been able to address the reviewer's concerns.